# Employee Satisfaction and Loyalty as a Part of Sustainable Human Resource Management in Postal Sector

## Mariana Strenitzerová * 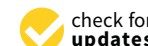 and Karol Achimský

Faculty of Operation and Economics of Transport and Communications, University of Žilina, Univerzitná 8215/1, 010 26 Žilina, Slovakia

* Correspondence: mariana.strenitzerova@fpedas.uniza.sk; Tel.: +421-41-513-3131

**Abstract:** The labor market situation in the postal and logistics sector has led to sustainable human resource management becoming increasingly important at the strategic level of each postal operator. This study proposes a new perspective of employee satisfaction assessment that not only quantifies total satisfaction but identifies job attributes and socio-demographic characteristics affecting employee satisfaction and loyalty as a key concern for sustainable human resource management. Findings of 1775 survey questionnaires of employees in Slovak Postal enterprise revealed that employee satisfaction is on average. The greatest dissatisfaction was related to the financial reward of employees and the employer's lack of interest in the views and attitudes of employees. The use of regression and correlation analysis pointed to the fact that not only their satisfaction, but also the situation on the labor market in the region, the age of the employee, the job position and the length of employment have a strong influence on employee loyalty.

**Keywords:** sustainability; sustainable human resource management; labor market in postal sector; employee satisfaction; employee loyalty

---

## 1. Introduction

The concept of sustainability is important for companies in the sector of postal services. Globalization, liberalization of postal market, the entry of competition into the postal market, and Internet of Things have led to significant changes across the functioning of postal companies. The changes also concerned human resource management. Sustainable human resource management is a typical cross-functional task that becomes more and more important at the strategies of postal companies.

Sustainable human resource management (HRM) can help postal companies to establish an attractive employer brand that can address the different needs and expectations of potential and existing employees, without compromising a consistent employer image, which can result in a sustained competitive advantage of the postal company. Sustainable HRM can help postal companies attract and retain high-quality employees, because by integrating sustainable HRM practices into the employee value proposition, they establish a unique, attractive employer brand. An interesting study in this area was carried out by App et al. [1] whose conclusions can also be applied to postal employers.

Postal companies become aware of their responsibilities towards employees and engage in sustainable HRM. In particular, they consider the creation of the best working conditions for safety, health, social background and continuous training of employees as their role. All these activities aim at the main goal of postal companies in the area of HRM—to increase the satisfaction and loyalty of their employees. However, are these activities effective and efficient? How do employees perceive them? Are they satisfied with this care by the employer? The aim of this study is to assess the

sustainable HRM practices from the perspective of postal company employees and its impact on their satisfaction and loyalty. We tried to include in our research the comprehensive activity of the Human Resource Division and the activities related to sustainable HRM. The postal sector suffers from a lack of workforce and therefore postal companies pay great attention to the satisfaction and loyalty of employees. Employer image, employee expectations, perceived quality of services of Human Resource Division, value perceived by employees are factors that significantly affect the employee satisfaction and employee loyalty to postal companies. The purpose of this study is to evaluate the impact of these factors (latent variables) on employee satisfaction and consequently evaluate the impact of employee satisfaction on their loyalty to the employer. Our research has drawn attention to the importance of relationships between employee expectations, perceived HR service quality, value perceived by employees and employee satisfaction. Our research has pointed out how demographic factors (age, length of employment for the postal provider, job classification, place of work—region, labor market in a region) influence satisfaction and loyalty of Slovak Post employees.

## 2. Theoretical Framework and Hypotheses Development

### 2.1. Sustainable Human Resource Management Literature

Companies willing to attract and retain human resources for running businesses in the future must change the prevailing situation where human resources are rather consumed than developed. In doing this, sustainable HRM has been introduced recently as a response to changes on societal levels, labor market, and employment relations. Sustainable HRM is seen as an extension of strategic HRM and presents a new approach to people management with the focus on long-term human resource development, regeneration, and renewal [2].

Researchers and other professionals unanimously agree that companies should become more sustainable, but this will not happen without the support of HRM [3]. Labbour and Santos [4], in their work, present the main contributions of HRM to develop sustainable organizations. The relationship between human resources and organizational sustainability involves some important aspects concerning management such as innovation, cultural diversity and the environment.

Kramar [5] examines the major features of sustainable HRM, some of the meanings given to sustainability and the relationship between sustainability and HRM. He emphasizes the need focus on the sustainable HRM contributing not only to financial outcomes, but also to the importance of human, social and ecological outcomes in terms of their contribution to business outcomes.

Esfahani et al. [6] investigate the main characteristics of a sustainable HRM in innovative organizations. The aim of their research was to find the relationship between psychological capital, HR flexibility and sustainable HRM in innovative organizations. They examined HRM in innovative organizations may benefit from psychological capital and the flexibility of human resources. They found that functional HR flexibility and optimism have maximum effect on HR sustainability.

Even family businesses, and small and medium enterprises (SME) need to think about their sustainability. Ping [7], in his research, presents some strategies and proposals for the innovation of HRM of a family business under the background of sustainable development. Liu and Yu [8] analyze HRM based on sustainable development in SMEs and bring forward some strategies and recommendations to improving HRM in SMEs.

Jarlstrom, Saru, Vanhala, illustrate the dimensions and broader responsibilities of sustainable HRM. Their research analyses how top managers construct sustainable HRM, its responsibility areas and how top managers identify and prioritize stakeholders in sustainable HRM [9].

Enterprise culture is one of the key factors of HRM. Wu et al. [10] confirmed that Harmony-Oriented Culture forms the basis for the sustainable development of enterprise management.

Kazlauskaite and Buciuniene [11] attach great importance to human resources and their management in the establishment of sustainable competitive advantage. They are convinced that human resources make the strategic value for an organization. Acquiring and sustaining of competitive

advantage necessitates unique, valuable, and inimitable employees and effective people management practices. According to Kucharcikova et al. [12] human capital management represents a modern concept of people management which also leads to the increase of performance and competitiveness of the enterprise within the context of sustainability. It is necessary measurement and assessment of the effectiveness of the utilization of human capital and effectiveness of investment in its development.

Realizing the sustainable development of economy is the most important strategic target of regional economic development. It is also decided by the human resource reasonable development and optimization. Wang and Wang [13] elaborated the relations between the HRM and the regional economic sustainable development. Sustainable employability policy can positively influence regional economic development. Van der Heijden [14] pays directed attention to sustainable employability policy as being part of strategic HRM.

Lis [15] focuses the relevance of corporate social responsibility for a sustainable HRM. Each aspect of Corporate Social Responsibility (CSR) has a specific effect on company's attraction and can be an effective tool to attract potential employees.

Macke and Genari [16], in their study, aim to analyze the state-of-the-art advancements of sustainable HRM and to identify key elements, trends and research gaps. They define four categories of studies. The first category comprises studies on sustainable leadership, based on individual and group power and is embedded in its principles, processes, practices and organizational values. The second category demonstrates the relationship among HRM, environmental sustainability and organizational performance. The third category considers the tensions and paradoxes between HRM practices and sustainability: on the one hand, HRM should focus on cost reduction and corporate profitability (in the short-term); on the other, their actions should provide long-term sustainability of organizational performance. The last category deals with the link between HRM and the social dimension of sustainability, especially with regard to organizational social responsibility and the company's relationship with its stakeholders.

Drawing on literature from a range of works linking sustainability and HRM and following the essence of corporate sustainability, Stankeviciute and Savaneviciene [2] propose 11 characteristics of sustainable HRM, namely: Long-term orientation, care of employees, care of environmental, profitability, employee participation and social dialogue, employee development, external partnership, flexibility, compliance beyond labor regulations, employee cooperation, fairness, and equality. All these characteristics affect the employee satisfaction and loyalty ratings, which are the subject matter of our research. The objective to succeed, ensure sustainability, remain competitive, and to increase business efficiency can only be ensured by satisfying the needs of employees by providing them with good working conditions [17–19]. The quality of work life positively and significantly influences employee job satisfaction, and employee loyalty [20]. Additionally, Al Mamun et al. [21], Cantele, and Zardini [22], Delmas, and Pekovic [23], Pintão et al. [24], and Dongho [25] referred to employee satisfaction as one of many challenges, since the productivity of employees was crucial to the company's success or failure.

Employees satisfied with their rewards and work environment do not have a need to leave the company; they are loyal. It is optimum when employees are satisfied with their conditions and work environment [26–28]. The same view was presented by Chang et al. [29], Chatterjee et al. [30], Gianni et al. [31], Roxas et al. [32], and Chandrasekar [33], who argued that the company needs to pay attention to creating a work environment that enhanced the satisfaction and motivation of employees in order to increase corporate sustainability performance.

Giovanis [34], explored the relationship between job satisfaction, employee loyalty and two types of flexible employment arrangements; teleworking and flexible timing. The author applied the Bayesian networks (BN) and directed acyclic graphs (DAGs) to confirm the causality between employment types explored and the outcomes of interest. Additionally, he proposed an instrumental variables (IV) approach based on the BN framework. The results support that a positive causal effect from these employment arrangements on job satisfaction and employee loyalty is present. The subject

of our research is also the relationship between job satisfaction, employee loyalty and job classification in condition of postal companies.

Kot-Radojewska and Timenko [35], examined the relationship between employee loyalty to the employer and the form of employment. The research results indicated that the employees who have an indefinite duration employment contract rated the degree of their own loyalty to the employer higher than people that have a fixed-term employment contract. This is an interesting finding that will be the subject of our research of dependence of employee loyalty on length of employment for Slovak Post.

Recently there has been a resurgence of interest in the analysis of job satisfaction variables. Job satisfaction is correlated with labor market behavior such as productivity, resignations and absenteeism. Gazioglu and Tansel [36], provide an empirical analysis of the determinants of job satisfaction in Britain. Four different measures of job satisfaction are used: satisfaction with influence over job; satisfaction with amount of pay; satisfaction with sense of achievement; and satisfaction with respect from supervisors. These four different measures of job satisfaction are related to a variety of personal and job characteristics. The main results can be summarized as follows: job satisfaction is U-shaped in relation to age; the better-educated are less satisfied relative to those with no or few qualifications; higher income produces higher levels of satisfaction; long working hours reduce satisfaction; satisfaction is lower in larger establishments; those who are in managerial, professional and clerical occupations are more satisfied than sales employees; those employees who had job training were more satisfied than those who had no training opportunities. These are interesting results that have led us to research how some of these characteristics affect employee satisfaction and loyalty in the postal sector. Baumgardt et al. [37] are also convinced that aspects to be considered to reinforce job satisfaction are age, years of practice, quality and quantity of cooperation.

It is generally believed that job satisfaction increases linearly with age. However, there are persuasive arguments, and some empirical evidence, that the relationship is U-shaped, declining from a moderate level in the early years of employment and then increasing steadily up to retirement. For overall job satisfaction, satisfaction with pay, and satisfaction with the work itself a strongly significant U-shape is observed. Clark et al. [38] thus provide strong evidence for a U-shaped relationship between age and job satisfaction. This is an interesting finding that has also been the subject of our research.

Sustainable HRM has a broad scope and all areas of its research can be applied to the postal sector and postal companies; these are among the major employers and attribute HRM great importance.

### 2.2. Sustainable Human Resource Management in Postal Sector

The postal sector demonstrates leadership and innovation in the field of global corporate sustainability. Significant international developments such as the adoption of the Paris Agreement and the rising prominence of the UN Sustainable Development Goals (SDGs) have signaled the need for more urgent sustainability action from the global business community. Accordingly, in 2019 International Post Corporation's (IPC) Environmental Measurement and Monitoring System (EMMS) program will be expanded beyond carbon management to encompass broader sustainability goals. IPC will set ambitious targets across a range of sustainability issues, designed to drive significant improvements and ultimately help the postal sector maintain its position as a sustainability leader [39].

Last year, the program's participants identified the five UN SDGs most material to the postal sector, which take action to mitigate climate change and cover issues such as ensuring decent working conditions, promoting innovation, building sustainable cities and communities, ensuring responsible production and consumption. The postal operators involved in the program have implemented many activities related to sustainable HRM. They mainly concerned decent working conditions and employee training.

The bpost, Belgium's largest postal service, has defined its wider Corporate Social Responsibility (CSR) strategy and its potential impact on the sustainability score, around three main pillars:

- People: care about employees and engage them with sustainably (employee health and safety, employee training and talent development, ethics and diversity, social dialogue);

- Planet: to strive to reduce its impact on the environment (green fleet, green buildings, waste management);
- Proximity: to engage with the community it serves (be close to the society: to community, to suppliers, to customers through its services).

The goal is the shared value creation: continuity of business, employee satisfaction and engagement, customer satisfaction.

The bpost plays a major role in society, and it strives to be an organization that its customers can trust. Embedding its CSR strategy and introducing innovative programs into its corporate operations and culture will help it reach its ambition of sustainable growth and demonstrating sustainability leadership.

In order to address the environmental impact of last-mile deliveries, Spanish operator Correos and Correos Express have joined forces to implement a sustainable delivery program. The Correos and Correos Express programs integrating efficiencies and new technologies underpin its commitment to the environment and capacity for innovation. Correos implemented route optimization systems and developed new delivery methods such as the CityPaq service, an automated collection point. These initiatives have engaged and brought satisfaction to employees, introduced more sustainable and cost-effective transport and reduced the environmental impact on local communities.

For Le Groupe La Poste, sustainability and innovation go hand in hand. France's postal operator has developed creative technological delivery solutions including a remote-controlled drone and an automated mail cart named Effibot. This autonomous cart accompanies the postman/postwoman during his/her delivery round and facilitates the work, stopping when they stop, circumventing obstacles and carefully avoiding pedestrians. The DPD group drone simplifies deliveries in secluded or difficult to access areas, and can facilitate deliveries in emergency situations, such as delivering medicine. These sustainable solutions are not only safer for employees and more efficient for customers. They are part of the new and innovative technologies that can help deliver environmental benefits and improve employees' working conditions.

Poste Italiane is empowering people through sustainability insights. Managing the impact of the mobility of people and goods is an ever increasing environmental and social challenge. Knowledge sharing, particularly around transportation and behavior, is a key element to empowering people to make choices that will have positive impacts for the community now and for generations to come. In 2017, Poste Italiane's corporate Mobility Management unit produced an e-book entitled Perché muoversi in modo sostenibile? (i.e., Why should we opt for a sustainable mobility?). This e-book illustrates a series of best practices and economic advantages linked to more environmentally sustainable lifestyles, aimed at promoting greater awareness and sustainable mobility amongst Poste Italiane's employees. The e-book is free and employees are encouraged to share these insights with their communities, thereby empowering a greater number of people to make more sustainable mobility choices. This knowledge sharing builds on Poste Italiane's 150-year history and aligns its reputation with progress, innovation and care for the community.

Driver training is another key way in which program's participants can achieve emissions reductions from own transport. Many of our participants have already introduced eco-driving initiatives and communications campaigns for their drivers and are already making progress in this area. Moreover, the participation of nine postal companies in IPC's fifth International Drivers' Challenge in April 2018 illustrates participants' ongoing commitment to reducing emissions from transport. Their participation also demonstrates their commitment to engaging employees in sustainability issues, a key part of effective carbon management. Through this event, which was hosted at the Estoril Racing track in Portugal, IPC emphasized the importance of economic and fuel-efficient driving behavior and demonstrated the benefits of investment in eco-driving initiatives.

Postal companies become aware of their responsibilities towards employees and engage in sustainable HRM. In particular, they consider the creation of the best working conditions for safety, health, social background and continuous training of employees as their role. Postal companies are

making considerable efforts towards continuous improvement in the area of occupational safety and health protection. Equally, in the implementation of education and in the consistent application of the principles of diversity and equal opportunities. An open communication culture, supporting employee engagement in these processes and activities, is a matter, of course, for postal companies. In this context, they realize that a motivating work environment is also important. All these activities aim at the main goal of postal companies in the area of HRM—to increase the satisfaction and loyalty of their employees. The postal sector suffers from a lack of workforce and therefore postal companies pay great attention to the satisfaction and loyalty of employees as a part of sustainable HRM. That is why Slovak Post, as a universal postal service provider and the second largest employer in Slovakia, welcomed cooperation in solving our research.

### 2.3. Labor Market Situation in the EU and Slovak Republic Postal Sector

The provision of postal services is of fundamental importance in terms of regional development, social inclusion and the economic and territorial cohesion of the EU. The postal sector has undergone significant changes in recent years as a result of technological advances and digitalization, and the modernization and diversification of postal services also has a strong impact on working conditions and employment in this sector. The liberalization of postal services as a process of removing obstacles to free enterprise in the postal market has had an impact on the development of the postal market in the Slovak Republic. However, its impact can be assessed from a variety of aspects. It influenced the activity of the universal service provider, contributed to the development of competition in the postal market, influenced the behavior and satisfaction of the customers of the postal market and also contributed to the development of the postal market. However, the liberalization of the postal market has also significantly affected the labor market situation in the postal sector and employment in the sector.

HR managers of postal operators find that getting and keeping quality staff is not as easy as it used to be in the past. Qualified staff in the area of postal services have the opportunity to use a much wider offer of work by postal operators. The need for sustainable, high-quality employment in this sector leads the postal operators to spend considerable investment to sustainable HRM, creating the HRM strategy and HR marketing activities.

Until the mid-1980s, post was considered to be part of the public service in all EU countries. In the 1990s, several state-owned postal organizations were privatized and became private by the organizations that provide postal services. The terminology also includes the term of a national operator or postal operator or "incumbent" to designate entities that "hold power" and "have a duty" resulting from their market position. At the same time, their position is historically conditioned, when in the past these entities represented monopolistic institutions in their countries most often in state ownership. The development of technologies and market also leads to their changes—the loss of monopoly and privatization, respectively change of ownership structure [40].

Competition is a healthy and desirable phenomenon in the whole economy, and the post is no exception, stimulating innovation—exploring new and existing customer service methods, including identifying potential customers and their needs. Success in a competitive environment is generally expressed not only by the benefits, but also by the quality of the HR activities and by the quality of staff (the level of management work, increasing moral awareness among employees, etc.).

Postal operators play an important role in the social policy of the state by creating employment opportunities. In many countries, operators operating in the postal markets are one of the largest employers, and secondarily, they create a high number of jobs in the electronics and downstream industries [40].

The European Commission collects data on postal services in cooperation with the National Regulatory Authorities of the Post (NRAs) of participating countries through the EU Postal Survey. Data are collected annually from the 33 countries that participate voluntarily in data collection, including the 28 EU Member States. The indicator for the number of employees of individual universal service

providers (Table 1) captures staff working exclusively in the field of postal services in the economic territory of the reference country [41]. The above statistics show that the number of employees of the universal service provider in Slovakia is decreasing. According to data of the Office for Regulation of Electronic Communications and Postal Services, currently 25 postal enterprises are registered in Slovakia. The statistics show that the average number of employees in postal services and courier services is up to 89.2% employees of universal service providers [42]. Employment with other postal operators (outside universal service providers) is the driving force behind overall employment growth in the postal sector. Employment rates for other postal service providers increased significantly between 2013 and 2015. Based on the results of the ERGP37 study, an increase of 29.8% for the number of persons employed by other postal service providers was recorded between 2008 and 2015 [43].

**Table 1.** The number of persons employed in the provision of postal services domestically (persons employed by universal service providers).

| Geo/Time | 2012 | 2013 | 2014 | 2015 | 2016 |
|---|---|---|---|---|---|
| Belgium | 29,922 | 28,747 | 27,479 | 24,703 | 24,850 |
| Bulgaria | 11,384 | 11,126 | 10,794 | 10,572 | 10,508 |
| Czechia | : | 62,678 | 59,610 | 58,890 | 58,345 |
| Denmark | 13,386 | 12,727 | 11,772 | 10,399 | 9314 |
| Germany | : | : | 148,518 | 148,669 | 146,826 |
| Estonia | 2049 | 2059 | 1640 | 1768 | 1804 |
| Ireland | 8729 | 8251 | 8006 | 7890 | 7803 |
| Greece | 8837 | 7977 | 7185 | 6859 | 8039 |
| Spain | : | : | 51,275 | 50,153 | 54,764 |
| France | : | : | : | 207,890 | 205,053 |
| Croatia | 8137 | 7394 | 7082 | 7912 | 8078 |
| Cyprus | 1224 | 1210 | 1690 | 1702 | 650 |
| Latvia | 4215 | 4095 | 4189 | 4236 | 4239 |
| Lithuania | 6482 | 6272 | 6019 | 5924 | 5783 |
| Luxembourg | 1402 | 1376 | 1415 | 1385 | 1385 |
| Hungary | 31,610 | 25,787 | 23,779 | 28,427 | 27,967 |
| Malta | : | : | : | : | : |
| Netherlands | : | 59,280 | 52,364 | 49,174 | 46,456 |
| Austria | : | 20,145 | : | : | : |
| Poland | 91,373 | 83,468 | 79,741 | : | : |
| Portugal | 11,715 | 11,043 | 10,765 | 10,812 | 10,881 |
| Romania | 32,887 | 27,451 | 26,784 | 24,518 | 24,139 |
| Slovenia | 5598 | 5352 | 5968 | 5831 | 5510 |
| Slovakia | 14,470 | 14,363 | 14,329 | 14,214 | 14,049 |
| Finland | 18,610 | 18,610 | 16,558 | 16,669 | 16,220 |
| Sweden | 23,912 | 23,721 | 21,853 | 21,462 | 20,272 |
| United Kingdom | 149,710 | 149,172 | 145,205 | 139,000 | 142,000 |
| Iceland | 1078 | 1053 | 1001 | 1093 | 1115 |
| Norway | 19,388 | 19,022 | 19,114 | 18,590 | 16,992 |
| Switzerland | 52,378 | 51,779 | 52,831 | 51,619 | : |
| Former Yugoslav Republic of Macedonia | 2288 | 2217 | 2306 | 2265 | 2343 |
| Serbia | 15,068 | 15,155 | 15,015 | 14,965 | 14,868 |

Source: https://ec.europa.eu/growth/sectors/postal-services/statistics_en.

The Committee on Employment and Social Affairs of the European Parliament notes that employment in universal service providers is declining as a result of declining volumes of letter shipments and current modernization and increasing automation. Replacing paper mail with digital tools, technological advances to enable citizens to access services and communications from home also led to a decline in the number of post offices and a decline in the number of postal service staff.

Even though the number of employees in the postal service sector is decreasing, the number of short-time employees, agency workers and the self-employed in this sector has risen. The trend is towards more flexible employment contracts, which in some cases can lead to precarious jobs without adequate protection for employees [44].

The average wage in the EU postal sector is about 13 euros per hour and increased by six percent in the period 2013–2016. Wages in the postal sector vary considerably across European regions, with Eastern Europe having the lowest average wage, around 3.5 euros per hour, and Western Europe the highest, around 23 euros per hour. North and South Europe show similar average wages ranging between € 12 and € 14 per hour.

Payroll mechanisms are influenced by various factors, from the general macroeconomic situation of the country to the financial position of the postal operator. Political factors also play a role, particularly in situations of conflicting policy objectives, maximizing financial returns from state-owned enterprises and maintaining a stable level of employment [43].

The liberalization of postal services has led to significant differences in some Member States as regards working conditions and wages offered by universal service providers and competing companies providing specific postal services. The evolution of the average nominal monthly salary of an employee in postal and courier services in the Slovak republic (SR) is below the average nominal wage throughout the economy (Figure 1) [45].

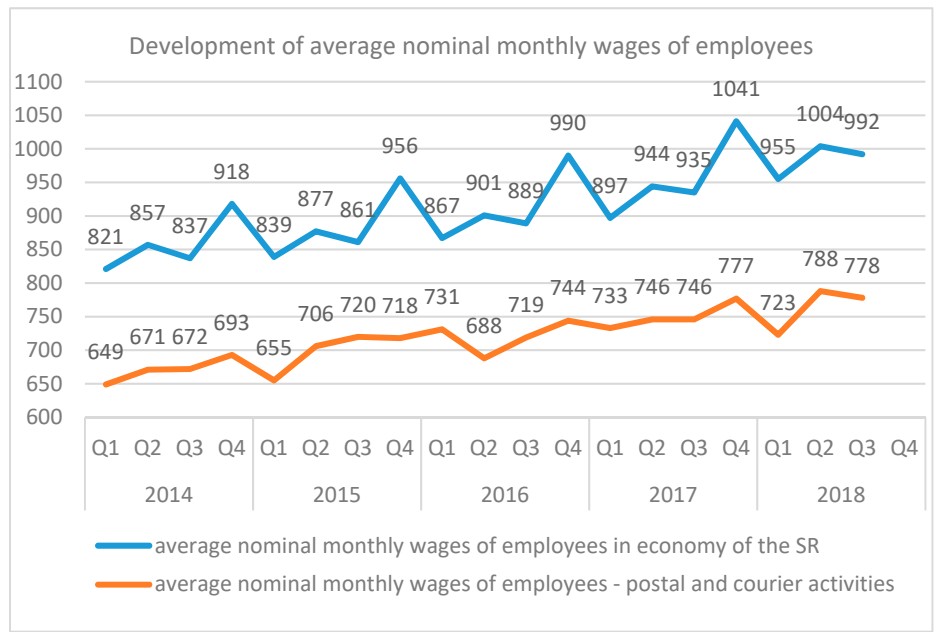

**Figure 1.** Development of average nominal monthly wages in sector of postal and courier activities. (Source: Statistical Office of the Slovak Republic. DATAcube. Created by author).

Average hourly wages at a universal service provider are often lower than average wages at national level. In most regions, wages in the transport and storage sector as a whole (including postal and courier services) are lower than average wages at national level.

The average monthly wage in postal and courier services of EUR 720 in the year 2016 was 86.4% of the average monthly wage in the transport, storage and postal services sector and was also the lowest average monthly wage within the sector [42].

Market opening and increased competition have forced national postal operators to modernize their wage structure. Such changes would, however, be more pronounced if collective labor contracts could be predefined and more flexible. Changes in minimum wages are an important driving force for salary development in the postal sector. However, minimum wage regimes differ in some major European economies. In France they are indexed by the consumer price index, while in Sweden there is no minimum wage regime. In many countries, the minimum wage is set by the state, which can be based on wages agreed on the market or negotiated with stakeholders.

Working condition rules and the role of the social partners also influenced the development of employment conditions in some postal markets. Social dialogue in the postal sector has played an important role in minimizing the negative impact of market changes on employment, for example through collective labor agreements. Employers and trade unions constructively work together to manage change in a socially responsible way. Coverage of trade unions and collective bargaining, however, shows the different levels of competence of trade unions in wage negotiations in individual countries.

Slovak Post (SP), as a universal service provider, is the second largest employer in Slovakia. In 2016, SP provided services by 13,446 employees in an average recalculated number. Of the total, women accounted for 82.50%. Compared to 2015, employment declined by 0.92%. Employees of SP were paid wages in the amount of 114,862 thousand €. The year-on-year growth rate of the average monthly wage of SP employees affected by the development of employment and wage earners reached 103.31% in 2016, it means the average monthly wage reached € 711.85 [46]. However, the average monthly wage is below the average in the national economy as well as in the sector of postal and courier services. The analysis of the labor market situation in the postal sector, the development of the number of employees and the development of the average wage in the postal sector have served as a basis for the assessment of employee satisfaction with the remuneration system and for the survey of employee loyalty.

## 3. Materials and Methods

Our research pointed out that the designed HRSI model can provide a better understanding of the complex relationships of the variables of employees' loyalty and employee satisfaction and their impact on sustainability of the postal provider. Factor analysis was used to summarize indicators (measurable variables) which identify the crucial factors. The regression and correlation analysis was used to examine the relationships between latent variables and measurable variables (indicators) which affect the satisfaction of employees and their loyalty to the employer. The regression and correlation analysis and association analysis was used to address the impact of demographic factors on employee satisfaction and loyalty.

### 3.1. Employee Job Satisfaction—Methodology

The academic literature has a long history of investigating employee job satisfaction. According to Spector, employee satisfaction is now a common concern among companies. Job satisfaction is a key factor in an employee's life, and thus job satisfaction is a stimulating topic to study [47]. Most academic research on this topic focuses on measuring and assessing job satisfaction [48–55]. When using the job elements comprehensive scoring method, scholars hold different views of the structure of job satisfaction. An often-used method is the Minnesota Satisfaction Questionnaire (MSQ), which classifies job satisfaction into four main aspects: work itself, interpersonal relationships on the job, reward and development [56]. Psychologist Smith advances Job Descriptive Index (JDI), which is comprised of five key dimensions, including satisfaction with: work itself, pay, supervision, opportunities for promotion, and co-workers. The JDI is designed to measure employees' satisfaction with their jobs. The JDI is a "facet" measure of job satisfaction, meaning that participants are asked to think about specific facets of their job and rate their satisfaction with those specific facets. The Job In General (JIG) is also designed

to measure employees' satisfaction with their jobs. The JIG is a measure of global satisfaction, meaning that participants are asked to think about how satisfied they are with their job in a broad, overall sense.

Many studies, however, provide a partial view of job satisfaction since they usually focus on the one-to-one relationship between an antecedent condition and job satisfaction, without taking a global view, to show how different factors simultaneously affect job satisfaction. This research posits that a combination of factors (e.g., employer image, employee expectations, perceived HR service quality, value perceived by employees) affects employee job satisfaction and loyalty. This study used a qualitative comparative analysis to explore the association between employee satisfaction, loyalty and the different socio-demographic factors and work attributes developed by postal employers.

As a suitable tool for this purpose, we created the HRSI (Human Resource Satisfaction Index) model (Figure 2), which can provide diagnostic of the complex relationships of the variables of employee satisfaction and their impact to loyalty of employee and sustainability of Slovak Postal provider. When creating the HRSI model, we based it on the general ECSI (European Customer Satisfaction Index) model, in which we considered Slovak Post employees as an internal customer of the Human Resource Division of the Slovak Post. The ECSI model monitors seven areas (latent variables) that have a determining impact on customer satisfaction, namely product/service/company image, customer expectations, customer perception, customer perceived value, customer satisfaction, customer complaints and customer loyalty. Measurable variables (indicators) within the monitored areas were determined after consultation with the management of Human Resource Division.

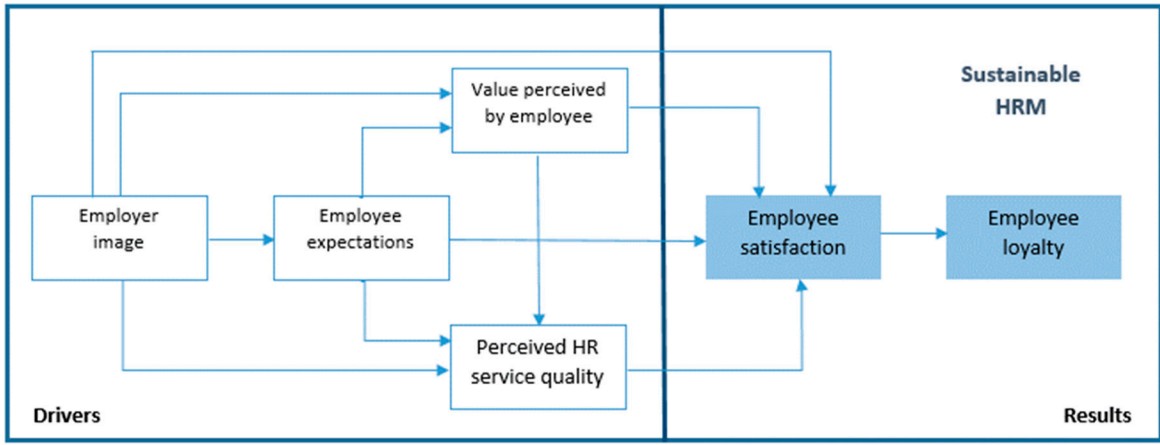

**Figure 2.** Human Resource Satisfaction Index (HRSI) model (created by author).

We designed the HRSI model to evaluate employee satisfaction and loyalty. The HRSI model represents one of the possible models to quantify employee satisfaction and loyalty. HRSI introduces six latent variables (employer image, employee expectations, perceived HR service quality, value perceived by employee, employee satisfaction, employee loyalty) corresponding to the areas of interest examined. Each of these latent variables is determined by measurable variables (22). This model is based on the use of a questionnaire survey among the employees of the postal company, which gives the primary input data [57].

Application of HRSI model within the research itself consisted of the following steps:

- Specification of measurable variables for HRSI model;
- Determining the importance of individual measurable variables;
- Converting measurable variables into questionnaire scale questions;
- Calculation of Human Resource Satisfaction index (HRSI).

The questionnaire consists of three parts. The first part deals with the socio-demographic characteristics of the participants in the research. The evaluation of variables: employer image (confidence, stability, flexibility and innovation, employer branding), meeting employee expectations

(in terms of workload, responsibility, rewarding, professional and career growth, organizational and personal provision, working environment, atmosphere and equipment at the workplace), perceived HR service quality (employee awareness, access to information, quality of communication, quality of superior-subordinate relationships, quality of teamwork, quality of education system), value perceived by employee (employee performance in relation to their workload, evaluation of social services), employee satisfaction (with job description, remuneration, professional and career growth), employee loyalty (fidelity, staff turnover) by respondents are in the second part of questionnaire. The level of satisfaction was expressed by one of the five levels based on the Likert scale (5 = very satisfied; 4 = satisfied; 3 = neutral; 2 = dissatisfied; 1 = very dissatisfied). The third part of questionnaire contains closed and open questions in order to better diagnostic the respondents' personal views concerning the monitored variables.

The application of the HRSI model, which includes the implementation of the employee satisfaction survey with the services of HR department of postal provider, has enabled us to find out what is the real satisfaction of Slovak Post's employees.

*3.2. Calculation of HRSI Index*

On the basis of the values obtained from employees through a questionnaire survey, satisfaction index according to relation 1 was calculated for each latent variable. We calculated the satisfaction index based on the general weighted arithmetic mean. Total satisfaction index values, calculated as the average of all indices, are expressed as a percentage. From these values, the overall value of the HRSI index was averaged [57].

$$I_i = \frac{\sum_{j=1}^{n} MV_{ij} \times w_j}{\frac{\sum_{j=1}^{n} w_j}{X}} \tag{1}$$

where:

| | |
|---|---|
| $I_i$ | satisfaction index of *i*-th employee |
| $MV_{ij}$ | value of *j*-th measurable variable for *i*-th employee |
| $n$ | amount of measurable variables |
| $X$ | scale extent (5) |
| $w_j$ | weight of *j*-th measurable variable |

The weights of measurable variables $w_j$ can be entered by the user or they are calculated as a covariance between the value of $x_{ji}$ and $y_i$, where $y_i$ is the sum of all measurable variables for the *j*-th latent variable that is divided by the number of these measurable variables [58].

$$w_j = \sum_{j=1}^{n} (MV_{ij} - P_j) \times (R_i - R_t) \tag{2}$$

where:

| | |
|---|---|
| $w_j$ | weight of *j*-th measurable variable |
| $MV_{ij}$ | value of *j*-th measurable variable for *i*-th employee |
| $n$ | amount of measurable variables |
| $P_j$ | average of *j*-th measurable variable for all employees |
| $R_i$ | average of values of all measurable variables for *i*-th employee |
| $R_t$ | average of values of all measurable variables for all employees. |

Latent variable is the result of the interaction of several measurable variables. Correlation analysis was used to investigate the dependencies between latent variables of HRSI model. Regression and correlation analysis, association analysis were used to assess the impact of demographic characteristics on employee satisfaction and loyalty.

For the purposes of calculating employee satisfaction indices for each latent variable was used Microsoft Excel®. XLStat is the leading data analysis and statistical solution for Microsoft Excel®,

which offers a wide variety of functions to enhance the analytical capabilities of Excel, making it the ideal tool for data analysis and statistics requirements. The statistical software program XLStat was used to analyze the values of latent and measurable variables and for calculating correlations and regressions of individual variables.

### 3.3. Profile of Respondents—Demographic Characteristic of the Sample

The subject of our research was employees of a significant Slovak postal provider. Slovak Post (SP) has a dominant position in Slovak postal market and sustainability have become the important part of its strategy. The strategy of SP is in line with The Postal Policy for 2021, which aims, in addition to economic growth, for a sustainable universal service and general improvement in the quality of life in society [59].

SP, as a universal service provider, is the second largest employer in Slovakia. In 2016, SP provided services by 13246 employees in an average recalculated number. The number of employees working at the SP represents 85% of the total number of employees working in the postal sector in Slovakia. Questionnaire survey of employee satisfaction was carried out at the postal provider SP, covering the entire company, in October 2018. In cooperation with the Human Resource Division of SP, the questionnaire was distributed electronically, by the Intranet. The survey was attended by 1775 respondents, representing a 13.4% return. The demographic information of respondents is presented in Table 2.

**Table 2.** Demographic characteristic of the sample.

| Demographic Variable | | Number of Respondents | Percentage of Respondents |
|---|---|---|---|
| *Age* | 18–30 years | 160 | 9 |
| | 31–40 years | 373 | 21 |
| | 41–50 years | 674 | 38 |
| | 51–60 years | 497 | 28 |
| | over 61 | 71 | 4 |
| *Length of employment for Slovak Post* | up to 1 year | 89 | 5 |
| | 1–5 years | 355 | 20 |
| | 6–10 years | 213 | 12 |
| | 11–20 years | 373 | 21 |
| | 21–30 years | 479 | 27 |
| | 31–40 years | 231 | 13 |
| | over 40 years | 35 | 2 |
| *Job classification* | operation | 1083 | 61 |
| | workers | 195 | 11 |
| | technical and economics | 408 | 23 |
| | other | 89 | 5 |
| *Place of work* | Banská Bystrica region | 373 | 15 |
| | Bratislava region | 266 | 21 |
| | Košice region | 213 | 12 |
| | Nitra region | 213 | 12 |
| | Prešov region | 231 | 13 |
| | Trenčín region | 160 | 9 |
| | Trnava region | 124 | 7 |
| | Žilina region | 195 | 11 |

The socio-demographic characteristics of the respondents show that most of the employees involved in the questionnaire survey were:

- In the age range: 41–50 years        674 respondents
- With a length of employment in SP: 21–30 years        479 respondents
- In the job classification: operation        1083 respondents
- Place of work in the region: Banská Bystrica        373 respondents

## 4. Results

Using the theoretical knowledge and results of employee satisfaction survey, we got information about determinants which have an impact on the sustainability of the postal provider (Slovak Post) and its HRM. The application of the HRSI model consisted of the identification of measurable variables determining the quality of the HR department's services and assessment of their relevance by employees (employee satisfaction). The application of HRSI model enabled better understanding of the potentially complex relationships between employee loyalty and employee satisfaction which have an impact on the corporate sustainability of SP in labor market in the sector of postal services.

### 4.1. Determining Measurable Variables and Its Assessment by Respondents

For the application of the HRSI model, it was first necessary to define the variables that we will monitor and measure at the postal provider. In Table 3, we present individual sets of measurable variables (22) that we assigned to each of the six areas (latent variables) that the HRSI model monitors. Based on results of questionnaire survey, we have calculated employee satisfaction index for each measurable and latent variable (Table 3, Figure 3).

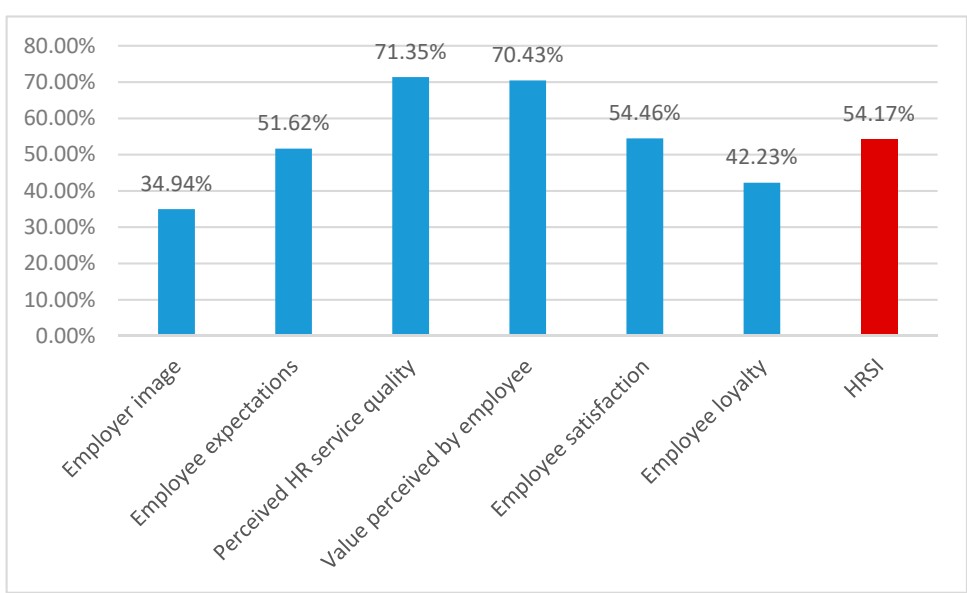

**Figure 3.** Human Resource Satisfaction Index (HRSI).

**Table 3.** Latent variables and measurable variables in HRSI model and their assessment.

| Latent Variables | | Measurable Variables/Indicators | Satisfaction Index (%) |
|---|---|---|---|
| *Employer image* | Img1 | Overall confidence in the business activity of postal provider and its HRM | 57.18 |
| | Img2 | Stability of the postal provider as an employer in the labor market (tradition, reputation, quality) | 32.23 |
| | Img3 | Flexibility and innovation of postal provider in relation to the needs and expectations of employees | 29.35 |
| | Img4 | Employer branding | 21.01 |
| *Employee expectations* | Exp1 | Fulfilling employee expectations in terms of workload (order of importance 3) | 59.76 |
| | Exp2 | Fulfilling employee expectations in terms of responsibility and decision-making powers (order of importance 2) | 32.23 |
| | Exp3 | Fulfilling employee expectations in employee rewarding (order of importance 1) | 33.51 |
| | Exp4 | Fulfilling employee expectations in professional and career growth (order of importance 6) | 70.10 |
| | Exp5 | Fulfilling the employee expectations in organizational and personal provision (order of importance 5) | 93.07 |
| | Exp6 | Fulfilling the employee expectations with regard to the working environment, the atmosphere and the equipment at the workplace (order of importance 4) | 75.53 |
| *Perceived HR service quality* | Qua1 | Employee awareness (access to information) | 82.48 |
| | Qua2 | Quality of workplace communication | 75.53 |
| | Qua3 | The quality of superior-subordinate relationships | 73.09 |
| | Qua4 | Quality of team work | 65.48 |
| | Qua5 | Quality of education system | 60.17 |
| *Value perceived by employee* | Val1 | Assessing employee performance in relation to their workload | 47.79 |
| | Val2 | Evaluation of social services provided to employees | 93.07 |
| *Employee satisfaction* | Sat1 | Satisfaction with the job description | 59.76 |
| | Sat2 | Satisfaction with employee remuneration | 33.51 |
| | Sat3 | Satisfaction with professional and career growth of employees | 70.10 |
| *Employee loyalty* | Loy1 | Fidelity to postal provider | 64.23 |
| | Loy2 | Staff turnover | 20.23 |

For determine the impact of demographic characteristics to employee satisfaction and loyalty was necessary to determine the satisfaction index for each demographic characteristic (Table 4).

**Table 4.** Assessment of measurable variables according to demographic characteristics.

| | | Img1 | Img2 | Img3 | Img4 | Exp1–6 | Qua1 | Qua2 | Qua3 | Qua4 | Qua5 | Val1 | Val2 | Sat1 | Sat2 | Sat3 | Loy1 | Loy2 |
|---|---|---|---|---|---|---|---|---|---|---|---|---|---|---|---|---|---|---|
| Age | 18–30 years | 66.25 | 45 | 21.25 | 28.13 | 40.38 | 85.25 | 78.26 | 73.71 | 65.01 | 60.04 | 45.83 | 93.18 | 54.04 | 35.48 | 66.15 | 47.2 | 34.16 |
| | 31–40 years | 63.27 | 39.41 | 26 | 26 | 54.25 | 84.75 | 76.75 | 75.86 | 66.49 | 61.63 | 49.29 | 94.43 | 59.35 | 36.41 | 67.24 | 55.97 | 23.34 |
| | 41–50 years | 52.22 | 28.04 | 28.93 | 19.44 | 49.76 | 81.81 | 74.6 | 72.55 | 64.62 | 58.43 | 44.1 | 92.66 | 59.28 | 31.59 | 69.61 | 59.88 | 21.26 |
| | 51–60 years | 55.73 | 28.77 | 35.21 | 18.71 | 51.01 | 80.72 | 74.42 | 71.44 | 65.47 | 61.7 | 51.45 | 92.64 | 62.18 | 32.75 | 73.61 | 79.12 | 13.32 |
| | over 61 | 61.97 | 29.58 | 28.17 | 9.86 | 53.54 | 82.95 | 79.8 | 73.74 | 69.7 | 58.08 | 54.44 | 92.42 | 62.5 | 37.5 | 74.24 | 83.33 | 15.15 |
| Length of employment for SP | up to 1 year | 77.53 | 43.82 | 26.97 | 41.57 | 56.64 | 85.75 | 78.85 | 77.78 | 69.53 | 65.59 | 48.75 | 92.47 | 58.87 | 40.32 | 71.51 | 59.13 | 31.18 |
| | 1–5 years | 57.46 | 43.66 | 24.22 | 29.3 | 54.38 | 85.37 | 76.27 | 74.16 | 66.47 | 60.9 | 47.05 | 93.95 | 56.84 | 35.45 | 67 | 53.03 | 26.22 |
| | 6–10 years | 52.58 | 32.86 | 26.76 | 15.02 | 51.41 | 84.91 | 74.21 | 74.37 | 65.72 | 62.11 | 47.27 | 92.92 | 57.55 | 34.67 | 68.63 | 57.55 | 22.64 |
| | 11–20 years | 56.57 | 29.49 | 30.29 | 22.79 | 51.03 | 81.71 | 75.88 | 73.16 | 65 | 59.56 | 49.17 | 93.16 | 59.74 | 33.88 | 68.03 | 62.89 | 18.42 |
| | 21–30 years | 57.62 | 24.43 | 31.1 | 15.66 | 48.8 | 80.31 | 74.31 | 71.67 | 63.61 | 58.19 | 44.75 | 92.29 | 60.57 | 30.36 | 70.78 | 68.13 | 17.5 |
| | 31–40 years | 54.98 | 32.48 | 35.93 | 15.15 | 53.2 | 81.5 | 76.21 | 72.69 | 66.67 | 60.5 | 54.35 | 93.83 | 64.98 | 33.59 | 77.09 | 81.5 | 17.18 |
| | over 40 years | 80 | 17.14 | 25.71 | 14.29 | 46.48 | 75 | 75.93 | 63.89 | 66.67 | 58.33 | 40.63 | 91.67 | 59.72 | 27.78 | 73.61 | 77.78 | 19.44 |
| Job classification | operation workers | 51.43 | 31.67 | 27.98 | 18.74 | 49.57 | 82.2 | 74.11 | 71.7 | 64.99 | 60.51 | 38.5 | 93.41 | 56.24 | 29.8 | 69.81 | 59.56 | 24.25 |
| | | 53.85 | 36.41 | 20.51 | 29.23 | 49.82 | 83.85 | 71.18 | 66.84 | 64.24 | 66.49 | 38.48 | 92.71 | 54.04 | 30.86 | 63.8 | 64.06 | 16.15 |
| | technical and economics | 75.25 | 34.07 | 39.22 | 23.04 | 59.11 | 83.13 | 81.72 | 79.65 | 67.49 | 56.49 | 79.07 | 92.31 | 73.45 | 45.41 | 74.88 | 79.16 | 9.92 |
| | other | 51.69 | 21.35 | 20.22 | 21.34 | 46.6 | 79.89 | 74.33 | 73.95 | 65.13 | 59 | 45.83 | 93.1 | 53.16 | 30.75 | 65.52 | 54.02 | 26.44 |
| Place of work—region | Banská Bystrica | 63 | 35.92 | 27.88 | 19.3 | 55.66 | 81.25 | 76.99 | 74.46 | 66.67 | 60.05 | 56.26 | 93.21 | 65.08 | 39.88 | 72.21 | 70.11 | 16.85 |
| | Bratislava | 49.62 | 22.93 | 26.69 | 14.66 | 49.96 | 81.95 | 75.68 | 74.9 | 62.16 | 53.28 | 53.64 | 93.44 | 61.2 | 34.27 | 70.08 | 62.55 | 25.87 |
| | Košice | 54.46 | 33.8 | 30.05 | 19.25 | 51.82 | 83.57 | 76.83 | 72.22 | 66.51 | 58.89 | 51.71 | 92.38 | 61.67 | 30.48 | 74.4 | 65.71 | 18.57 |
| | Nitra | 39.23 | 36.62 | 34.27 | 33.8 | 51.35 | 83.26 | 74.31 | 71.1 | 65.44 | 65.14 | 44.44 | 94.5 | 58.26 | 29.7 | 72.02 | 59.63 | 20.18 |
| | Prešov | 59.58 | 31.17 | 28.57 | 21.65 | 50.64 | 82.54 | 75 | 72.99 | 67.82 | 62.07 | 41.51 | 92.67 | 57.11 | 33.41 | 68.43 | 68.53 | 12.07 |
| | Trenčín | 57.5 | 33.13 | 26.25 | 18.75 | 48.85 | 82.41 | 73.46 | 72.43 | 64.4 | 58.23 | 38.26 | 91.96 | 55.4 | 27.93 | 66.67 | 60.49 | 19.75 |
| | Trnava | 49.19 | 31.45 | 27.41 | 23.39 | 51.34 | 85.25 | 75.14 | 72.95 | 64.75 | 68.31 | 50.15 | 90.98 | 56.97 | 36.07 | 69.26 | 61.48 | 27.87 |
| | Žilina | 51.78 | 32.31 | 34.36 | 18.97 | 50.11 | 81.74 | 75.16 | 72.06 | 65.2 | 59.64 | 38.43 | 94.12 | 56.13 | 31.25 | 64.95 | 58.82 | 25.98 |
| Summary | | **57.18** | **32.23** | **29.35** | **21.01** | **51.62** | **82.48** | **75.53** | **73.09** | **65.48** | **60.17** | **47.79** | **93.07** | **59.76** | **33.51** | **70.1** | **64.23** | **20.23** |

### 4.1.1. Employer Image

The overall satisfaction rate with the employer image is only 34.94%. Respondents appreciate their employer's reliability (32.17% of respondents), customer orientation (29.35% of respondents), credibility (25.01% of respondents) and prospect (19.32% of respondents). A relatively high number of respondents perceive SP as an entrepreneurial successful employer (12.90% of respondents) supporting professional development of employees (11.67% of respondents).

On the other hand, up to 46.31% of respondents perceive their employer as not interested in the views and attitudes of employees. This opinion was expressed mainly by respondents in the age group of 41–50 years (50.30% of respondents), with employment in SP more than 40 years (57.14% of respondents), with working position of operation (50.97% of respondents) and workers in the Trenčín region. It follows that SP should focus more on its personnel policy and build the employer's brand.

### 4.1.2. Employee Expectations

The overall satisfaction rate with the expectations of employees is 51.62%. When meeting the expectations of employees, we based it on the employees' priorities. According to the order of importance, the most important for employees is the wage and financial valuation of their work, followed by certainty of permanent employment, working conditions, interpersonal relationships at the workplace, benefits provided by the employer and, last but not least, training supporting professional and personal development. However, the order differs in individual employee segments. While in the age category of respondents 18–30 years working conditions are important in addition to wages and financial valuation of work, in other respondents' age categories it is a certainty of permanent employment. The same applies to the length of employment. Respondents working for SP less than 1 year appreciate working conditions in addition to wages and financial valuation. Other respondents prefer except for wage and financial valuation, a certainty of permanent employment.

### 4.1.3. Perceived HR Service Quality

The overall rate of satisfaction with the perceived quality of service provided by Human Resource Division is 71.35%.

*Employee awareness, access to information:* The overall satisfaction rate with awareness by direct supervisor is high (82.48%). Up to 81.92% of respondents rated it as positive. Only 1.01% of respondents showed great dissatisfaction. Good communication and awareness is the result of well-functioning relationships between the manager and employees, which were also positively evaluated in 89.01% of respondents.

SP uses various information channels to inform about SP events, about the implementation of its activities and programs. In-house magazine Poštové Zvesti is considered to be the main information channel providing information concerning personnel activities (e.g., implementation of social policy, social program, benefits provided to employees). Up to 60.11% of respondents use this as a base feed, followed by working meetings (53.58% of respondents) and intranet (53.30% of respondents). However, access to electronic information is limited for some employees, as evidenced by the low percentage of respondents using the electronic version of the eZvesti in-house magazine, e-mail communication and the SP website. Only 9.46% of respondents are looking for information on the employer's website. It is necessary to reflect on the design and recency of the website, as the intranet and the employer's website are the most widely used information channels of most companies.

*Quality of workplace communications and relationships:* Overall satisfaction rate with quality of workplace communications and relationships is 75.53%. Informal relationships in the workplace and a good team are very important to employees and are considered satisfactory to 94.82% of respondents. The dissatisfaction was particularly apparent in the operating staff (5.86%). They see the biggest shortcomings as being a bad atmosphere (slander, intrigue, envy, rivalry, etc.) resulting from the feeling

of overloading, the opaque division of tasks and the requirements for individual plans, staff shortages and frequent representation.

*Quality of relationships superior-subordinate:* In terms of their relationship to a superior manager, most respondents rate highly, and overall satisfaction is 73.09%. Following this, 89.01% of respondents consider it satisfactory or above standard. Satisfactory relationships with their superiors were the largest, ranging from 44.67–66.67%. The percentage of respondents who rated relations with their senior manager as above-standard was in the range of 13.89–47.64%. Above-standard relationships with the superior were mainly assessed by respondents in the 31–40 age group (37.67% of respondents), with employment up to 1 year (36.56% of respondents), with technical-economic classification (47.64% of respondents) and in respondents from the Bratislava region (36.68% of respondents). Good assessment of managers may also be due to the fact that most executives already have well-established work ties and are well-oriented in the work environment. Respondents attach great importance to the manager's personality, which was reflected in their assessment.

The percentage of respondents who rated relationships with senior managers as bad was in the range of 0.88–3.65%. These bad relationships occurred mainly among respondents over 60 years of age (3.03% of respondents), over 40 years of employment in SP (2.78% of respondents), job classification—workers (3.65% of respondents) and respondents working in the Nitra region (3.21% of respondents). In the case of negative assessments, respondents were dissatisfied mainly with the way managers were managed.

*Quality of team work:* 89.07% of respondents evaluate the cooperation with colleagues on other sections of the company satisfactorily, and the overall satisfaction rate is 65.48%. The largest proportion of respondents dissatisfied with cooperation with other departments is the category of operation employees. The biggest shortcomings were the poor organization of work on individual organizational units, reluctance, lack of communication, and lack of interest in their problems.

*Quality of the education system:* In terms of perceived quality of HR services, the lowest satisfaction was found in the quality of education (60.17%). The extent and quality of education provided by the employer is considered to be insufficient, mainly by the respondents of category technical and economics (38.71% of respondents), in the age category more than 60 years (37.88% of respondents), working in SP more than 40 years (44.44% respondents), from the Bratislava region (43.24% of respondents).

### 4.1.4. Value Perceived by Employee

The overall satisfaction rate with value perceived by employees is 70.43%.

*Assessing employee performance in relation to their workload:* Overall satisfaction rate with the performance evaluation system is low (47.79%). Thus, 48.62% of respondents say that the indicators set are not in line with their job description or they are unable to achieve them, which can cause employee frustration. Following this, 10.25% of respondents say they have no indicators at all. The greatest dissatisfaction was found in respondents of category operation (61.85% of respondents).

In assessing the quality of the evaluation system, we also focused on the regularity and frequency of the evaluation process. Most respondents said they are rated once a year (41.35% of respondents), 36.90% of respondents monthly. Up to 8.79% of respondents said that their senior manager do not evaluate at all and 4.90% of respondents do not know when they are evaluated. Mostly they are employees from the workers category. This suggests that these employees do not know the staff evaluation system and are not sufficiently informed about the results of their evaluation.

*Evaluation of social services (benefits) provided to employees:* Although in terms of importance the benefits provided by the employer are in fifth place, the overall satisfaction with the offer of benefits was the highest compared to other monitored indicators (93.07%). Respondents said that the most beneficial reward for them is a financial contribution to supplementary pension saving or life insurance. Followed by the following: the financial allowance for meals and the weekly working time of 37.5 h. Financial contributions to support the family is the least beneficial for respondents. Furthermore, 6.93% of respondents do not consider any social measures and benefits to be beneficial.

### 4.1.5. Employee Satisfaction

The overall employee satisfaction rate is 54.46%.

*Satisfaction with the job description:* The overall satisfaction rate with the job content is 59.76%. The highest satisfaction was found in the age category of respondents over 60 (62.50%), respondents in employment in SP 31–40 years (64.98%), respondents in technical-economic classification (73.45%), and in Košice region (61.67%).

*Satisfaction with employee remuneration:* Although wages and financial valuation are ranked first in terms of importance, overall satisfaction with financial valuation is only 33.51%. This discrepancy between the perception of the importance of wages and satisfaction with financial valuation can be a source of employee demotivation. Up to 83.21% of respondents consider financial valuation unsatisfactory. The most dissatisfied respondents are from the operation category (87.56% of respondents), and the overall satisfaction rate is only 29.80%. The lowest level of satisfaction was found in respondents from the Trenčín region (27.93%).

*Satisfaction with professional and career growth of employees:* The overall satisfaction rate with professional and career growth is 70.10%. The highest level of satisfaction was found in respondents of technical-economic classification (74.88%), with employment in SP 31–40 years (77.09%). This is a category of employees where rapid career growth is allowed, and a category of employees who already have sufficient experience obtained in SP.

### 4.1.6. Employee Loyalty

The results of loyalty and fluctuation tendencies are relatively favorable. Up to 64.23% of respondents do not consider changing their employer. Employee loyalty is directly proportional to the age structure of employees, i.e., the older the employee, the more loyal. From the point of view of the length of employment at SP, there are similar tendencies, the highest loyalty was found for employees with employment in SP 30–40 years (81.5% of respondents). High loyalty was reflected in the respondents of technical and economics job classification (79.16% of respondents). In terms of the place of work, the highest loyalty was found in respondents working in Banská Bystrica region (70.11% of respondents), Prešov region (68.53% of respondents) and Košice region (65.71% of respondents). This is also due to the fact that these are regions where unemployment is highest in Slovakia.

Considering employees who are considering a change of employer within 1 year, the largest group consists of employees aged 18–30 (34.16% of respondents), with employment up to 1 year (31.18% of respondents); job classification—operation (26.44% of respondents), with a place of work in Trnava region (27.87% of respondents), Žilina region (25.98% of respondents) and Bratislava region (25.87% of respondents). These are regions where is the lowest unemployment rate in Slovakia and the labor market offers much more interesting job opportunities.

Figure 3 shows employee satisfaction index of six latent variables, and total Human Resource Satisfaction Index (HRSI). The results show that the total satisfaction of SP employees is only 54.17%. The highest satisfaction was in the perceived HR service quality (71.43%). The Human Resource Division is making great efforts to ensure quality personnel services. Personnel activities are constantly innovated, taking into account the results of employee satisfaction surveys. The lowest satisfaction was in the area of the employer image (34.94%). Every postal company, knowingly or unconsciously, creates its own image, often independently of its own will. It is necessary that the postal company take care of its image, build it up and strengthen it. Due to increasing competition in the postal market, and especially after the liberalization process, the concept of image becomes an important tool for positive perception by the individual or society [60]. It is necessary for SP to focus its attention on building an employer brand. It is important to know what not only current but also potential employees think about SP. The attractive employer brand fulfills the different needs and expectations of potential and existing employees and can be a sustained competitive advantage of the SP in the postal market.

### 4.2. Regression and Correlation Analysis of Latent Variables in HRSI Model

The questionnaire survey and calculation of overall satisfaction indices and satisfaction indices according to individual demographic characteristics were used as input data for dependence research of demographic characteristics to employee satisfaction and loyalty. The correlation, regression and association results have supported most of the expectations and the hypotheses in this study.

Figure 4 shows the results of correlations between individual latent variables of HRSI model. It is clear from the results that employee-perceived value has the highest impact on employee satisfaction. The coefficient 0.90 shows an almost perfect correlation. This means that the performance appraisal of the employees in relation to their workload and evaluation of the social services and benefits provided to employees significantly affects the overall employee satisfaction. However, satisfaction with performance appraisal of the employees in relation to their workload was very low (satisfaction index 47.79%). The satisfaction with the offer of social services and benefits provided to employees was much higher (93.07%). We expected that the greatest impact on employee satisfaction has the perceived quality of HR services (E1). Based on the value of the correlation coefficient (0.44 < 0.90) our expectation E1 has not been confirmed

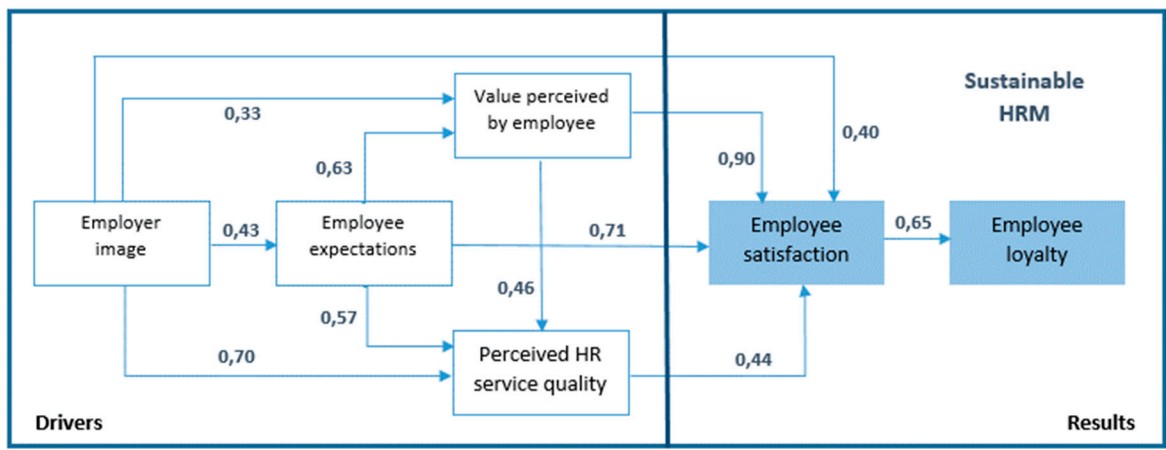

**Figure 4.** Structural model—results.

A large correlation affecting employee satisfaction was found in employee expectations (0.71). Fulfilling employee expectations in employee rewards, in terms of responsibility and decision-making powers, and in terms of workload, which has a significant priority for employees and fundamentally affect their satisfaction, was evaluated as being very low (satisfaction index 32.23-59.76%). Perceived HR service quality (0.44) and employer image (0.40) represent a moderate correlation to employee satisfaction.

We found a large correlation between employee satisfaction and loyalty (0.65). Based on this fact, our expectation E2 has been confirmed. As part of the latent variable "Loyalty", we also examined the employee turnover indicator. An indirect correlation was found between satisfaction and staff turnover (with the increase in satisfaction, the employee turnover decreases), showing a mean value (−0.43).

### 4.3. Impact of Demographic Characteristics to Employee Satisfaction and Loyalty

The aim of our research was to investigate the impact of age, length of employment for SP, job classification and place of work on employee satisfaction and loyalty. Regression and correlation analysis (for quantitative characteristics: age, length of employment) and association analysis (for qualitative characteristics: job classification, place of work—region) were used to investigate dependency.

Figure 5 presents the dependence of satisfaction indicators (Sat1, Sat2, Sat3) from the employee's age. The relationship between employee age and satisfaction with work load (Sat1) has a slightly increasing linear course (0.1235), while the determination coefficient value ($R^2$ = 0.8285) points to a

strong relationship. The satisfaction of the addressed employees (respondents) with their workload increases linearly with the employee's age; the expectation E3 has been confirmed. The relationship between the employee's age and satisfaction with the employee remuneration (Sat2) has a polynomial course, with the satisfaction of the employees of the 41–50 age group decreasing, while the satisfaction of other employees is increasing. The determination coefficient ($R^2$ = 0.9882) points to a very strong relationship. The satisfaction of the respondents with their remuneration is polynomially changing with the employee's age; the expectation E4 has not been confirmed. The relationship between employee age and satisfaction with professional and career growth (Sat3) has a slightly increasing linear course (0.25), although higher than the Sat1 indicator. The determination indicator ($R^2$ = 0.9397) points to a very strong relationship. The satisfaction of the respondents with professional and career growth of employees increases linearly with the employee's age; the expectation E5 has been confirmed.

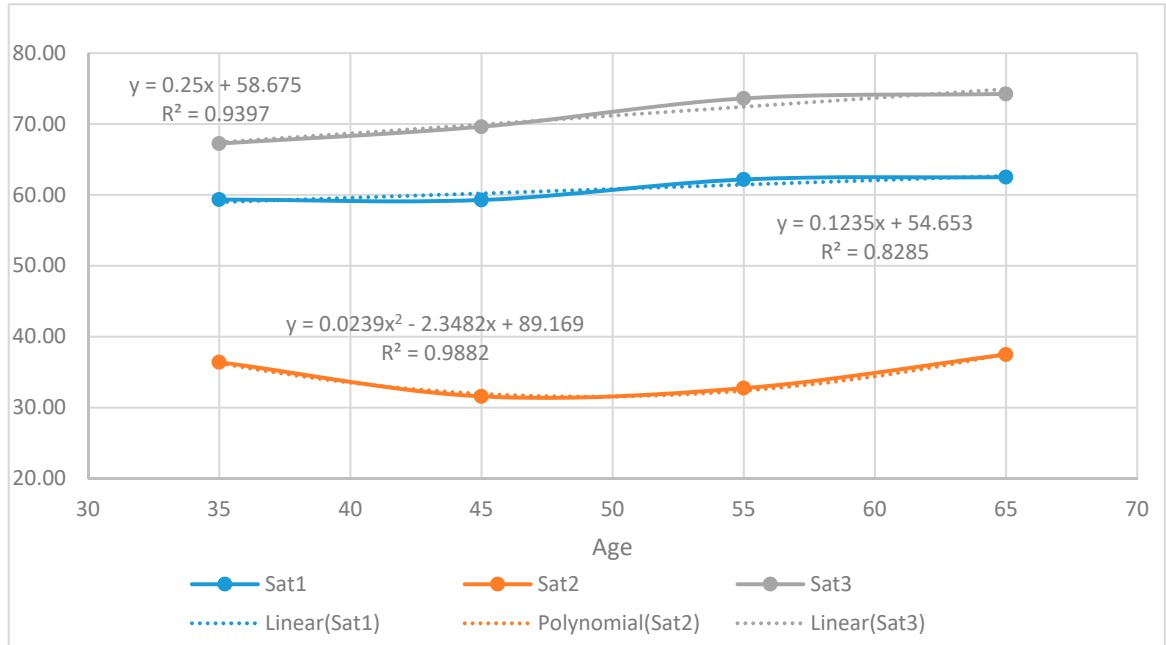

**Figure 5.** Dependence of satisfaction indicators (Sat1, Sat2, Sat3) on employee age.

Figure 6 presents the dependence of satisfaction indicators (Sat1, Sat2, Sat3) on the length of employment for SP. The relationship between length of employment and satisfaction with work load (Sat1) has a slightly increasing linear course (0.1779), the determination coefficient value ($R^2$ = 0.9394) points to a strong relationship. Satisfaction of the respondents with their work load increases linearly with length of employment for SP; the expectation E6 has been confirmed. The relationship between length of employment and satisfaction with employee remuneration (Sat2) has a slightly decreasing linear course (−0.2259), with the number of years worked for SP, the satisfaction decreases. The determination coefficient ($R^2$ = 0.9355) points to a very strong relationship. Satisfaction of the respondents with their remuneration decreases linearly with length of employment for SP; the expectation E7 has not been confirmed. The relationship between length of employment and satisfaction with professional and career growth of employees (Sat3) has a slightly increasing linear course (0.1472). The determination coefficient ($R^2$ = 0.7876) points to a strong relationship. Satisfaction of the respondent with their professional and career growth increases linearly with length of employment for SP; the expectation E8 has been confirmed.

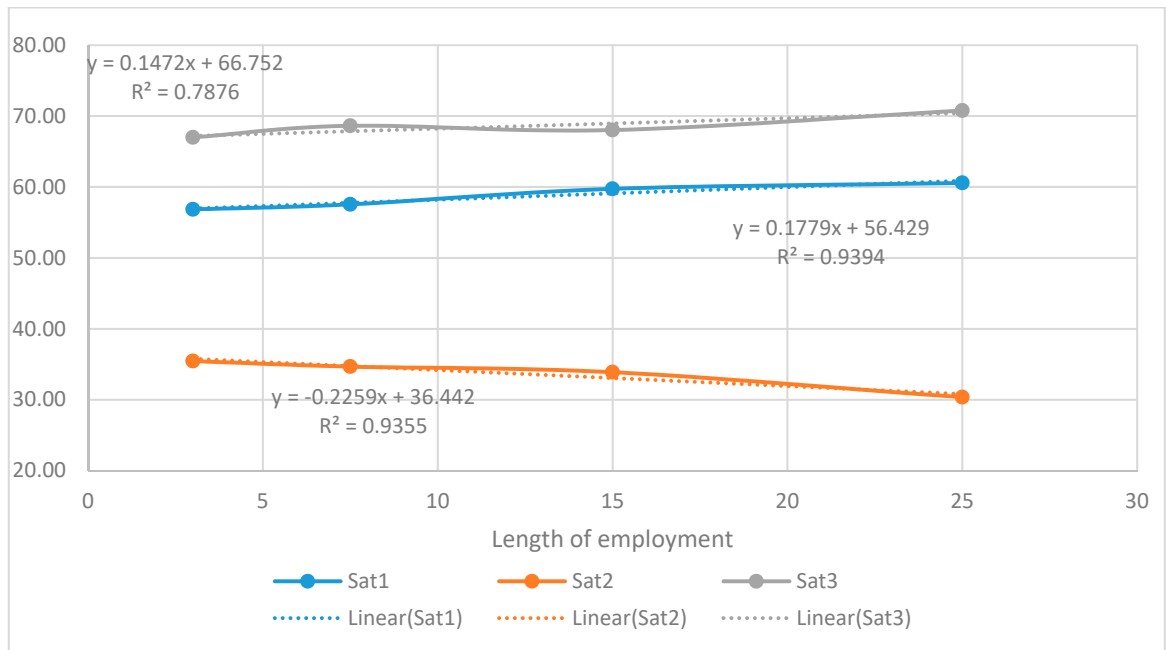

**Figure 6.** Dependence of satisfaction indicators (Sat1, Sat2, Sat3) on length of employment for SP.

Figure 7 presents the dependence of the loyalty indicators (Loy1, Loy2) from the employee's age. The relationship between employee's age and loyalty to SP (Loy1) has a slightly increasing linear course (0.9541), the determination coefficient value ($R^2$ = 0.9483) points to a very strong relationship. With an increase in the age of respondents, employee loyalty is increasing linearly; the expectation E9 has been confirmed. The relationship between employee age and staff turnover (Lyo2) has a slightly decreasing linear course (−0.4804), the determination coefficient value ($R^2$ = 0.8518) indicates a strong relationship, although is lower than the determination coefficient value of fidelity. With an increase in the age of respondents, employee turnover is decreasing linearly; the expectation E10 has been confirmed.

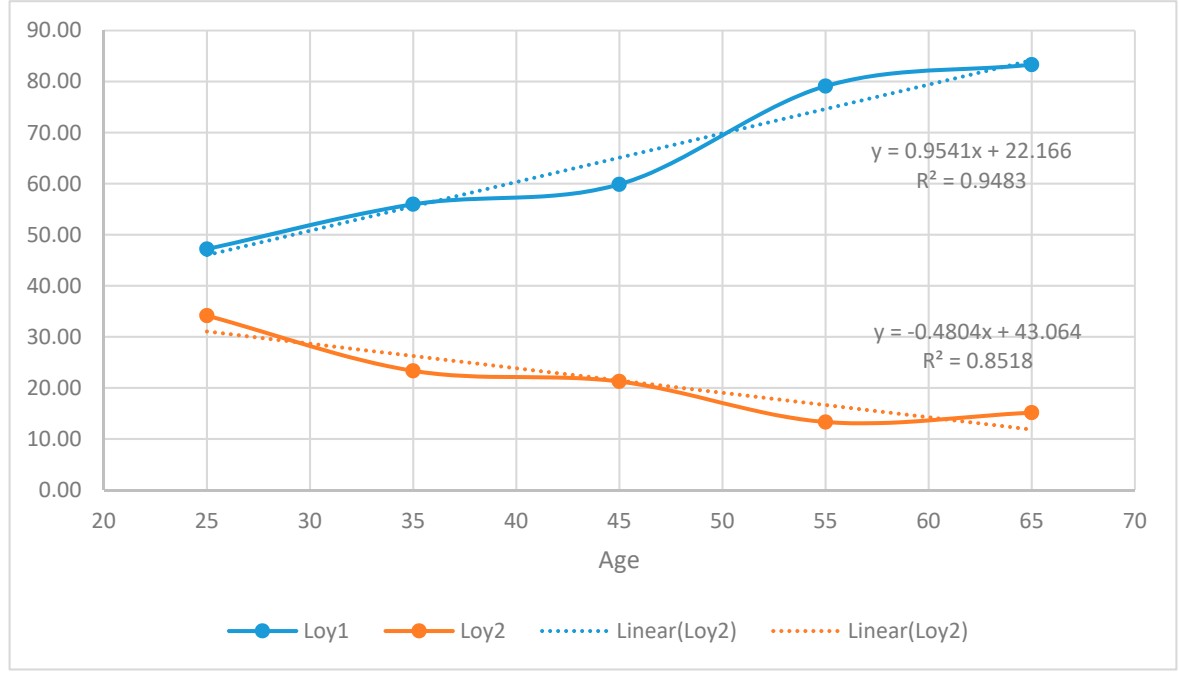

**Figure 7.** Dependence of loyalty indicators (Loy1, Loy2) on employee age.

Figure 8 presents the dependence of loyalty indicators (Loy1, Loy2) on the length of employment for SP. The relationship between employment length and loyalty to SP (Loy1) has a slightly increasing linear progression (0.5897), the determination coefficient value ($R^2 = 0.8772$) points to a strong relationship. With the increase in employment in SP, respondents' loyalty is increasing linearly; the expectation E11 has been confirmed. The relationship between length of employment and staff turnover (Lyo2) has a polynomial course; for employees with employment length below 40 years, the fluctuation is decreasing, but for employees with employment length over 40 years, the fluctuation increases. The value of the determination coefficient ($R^2 = 0.951$) points to a very strong relationship. With the increase in employment in SP, respondents' turnover decreases polynomially; the expectation E12 has not been confirmed.

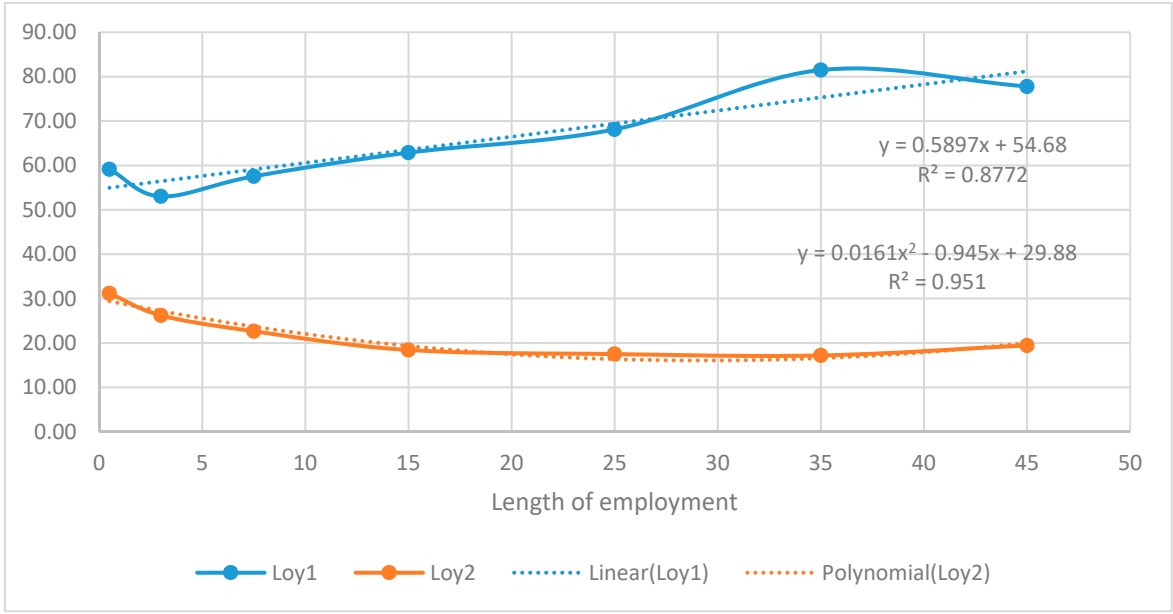

**Figure 8.** Dependence of loyalty indicators (Loy1, Loy2) on length of employment for SP.

The association analysis was used to explore dependencies between satisfaction levels and job classification. An examination of association confirmed the dependence between satisfaction with workload (Sat1) and job classification. However, the level of relationship is low (Pearson coefficient C = 0.30). Hypothesis H1 has been confirmed. An examination of association confirmed the dependence between satisfaction with employee remuneration (Sat2) and job classification. The level of relationship is lower (Pearson coefficient C = 0.24). Hypothesis H2 has been confirmed. An examination of association confirmed the dependence between satisfaction with professional and career growth of employees (Sat3) and job classification. The level of relationship is the lowest (Pearson coefficient C = 0.17). Hypothesis H3 has been confirmed.

The association analysis was used to explore dependencies between satisfaction levels and place of work (region). An examination of association confirmed the dependence between satisfaction with workload (Sat1) and place of work. However, the level of relationship is low (Pearson coefficient C = 0.17). Hypothesis H4 has been confirmed. An examination of association confirmed the dependence between satisfaction with employee remuneration (Sat2) and place of work. The level of relationship is higher (Pearson coefficient C = 0.18). Hypothesis H5 has been confirmed. An examination of association confirmed the dependence between satisfaction with professional and career growth of employees (Sat3) and place of work. The level of relationship is the highest (Pearson coefficient C = 0.20). Hypothesis H6 has been confirmed.

The association analysis was used to explore dependencies between loyalty indicators and job classification. An examination of association confirmed the independence between the level of

loyalty and job classification. Hypothesis H7 has not been confirmed. An examination of association confirmed the independence between staff turnover (Loy2) and job classification. Hypothesis H8 has not been confirmed.

The association analysis was used to explore dependencies between loyalty indicators and place of work (region). An examination of association confirmed the independence between employee loyalty (Loy1) and place of work. Hypothesis H9 has not been confirmed. An examination of association confirmed the independence between staff turnover (Loy2) and place of work. Hypothesis H10 has not been confirmed.

## 5. Discussion

Many companies are struggling with existential concerns, and disregard the needs of their employees. However, the interest in employees, their self-actualization, mutual expectations, and healthy labor relations are the preconditions for commitment, satisfaction, loyalty, and strong work ethic of employees and, thus, integral to the success of the entire company [17].

The purpose of this study was to review the satisfaction of employees in postal companies and consequently evaluate the impact of employee satisfaction on their loyalty to the employer. The aspects to be considered to reinforce job satisfaction are age, years of practice, job position and place of work. We investigated how these aspects affect employee satisfaction in the postal sector and compared our results with results of other authors' research.

The results of our research show that the total satisfaction of SP employees is only 54.17%. It is not the best result, although we can see significant differences in satisfaction from the point of view of employees' individual categories. In terms of age, the most satisfied category is employees over 61 years of age. From the point of view of the length of employment, the most satisfied are employees with an age of 31–40 years for Slovak Post. Satisfaction also varies greatly depending on job position. Above-average satisfaction was found in technical and economics employees. The situation on the labor market, the development of employment in the regions also have an impact on the satisfaction and loyalty of Slovak Post employees. The highest satisfaction was found in the Banská Bystrica region. The result was probably influenced by the fact that the company is headquartered in Banská Bystrica.

The development of loyalty is similar to the development of satisfaction. High loyalty was found in employees over 61 years of age, with 31–40 years of work experience on Slovak Post, and technical and economics employees in Banská Bystrica region. On the contrary, a very high fluctuation of employees was found in young employees, aged 18–30, with a practice less than 1 year for Slovak Post, with job classification "other" (subsidiary staff) in Trnava region.

It would be advisable for management of Human Resource Divisions to consider these disproportions when choosing appropriate motivation tools that support the satisfaction of Slovak Post employees (new option of employee benefits—Cafeteria system).

According to Table 5, employee age has the greatest impact on satisfaction with employee remuneration ($R^2 = 0.9882$); length of employment for SP has the greatest impact on employee turnover ($R^2 = 0.951$). Job classification has the greatest impact on satisfaction with work load (C = 0.30); place of work—region has the greatest impact on satisfaction with professional and career growth (C = 0.20). The impact of qualitative characteristics (job classification, place of work—region) to loyalty indicators (Loy1, Loy2) has not been confirmed.

**Table 5.** Dependence of satisfaction and loyalty indicators from demographic characteristics.

| Characteristic/Indicator | Sat1 | Sat2 | Sat3 | Loy1 | Loy2 | Coefficient |
|---|---|---|---|---|---|---|
| *Age* | 0.8285 | 0.9882 | 0.9397 | 0.9483 | 0.8518 | Determination coefficient $R^2$ |
| *Length of employment for SP* | 0.9394 | 0.9355 | 0.7876 | 0.8772 | 0.9510 | |
| *Job classification* | 0.30 | 0.24 | 0.17 | - | - | Pearson coefficient C |
| *Place of work—region* | 0.17 | 0.18 | 0.20 | - | - | |

According to Figure 9, satisfaction with work load is most influenced by length of employment for SP and by job classification of employees. Satisfaction with employee remuneration is most influenced by age of employees and by job classification of employees. Satisfaction with professional and career growth is most influenced by age of employees and place of work—region. Loyalty of employees is most influenced by age of employees, staff turnover is most influenced by length of employment for SP.

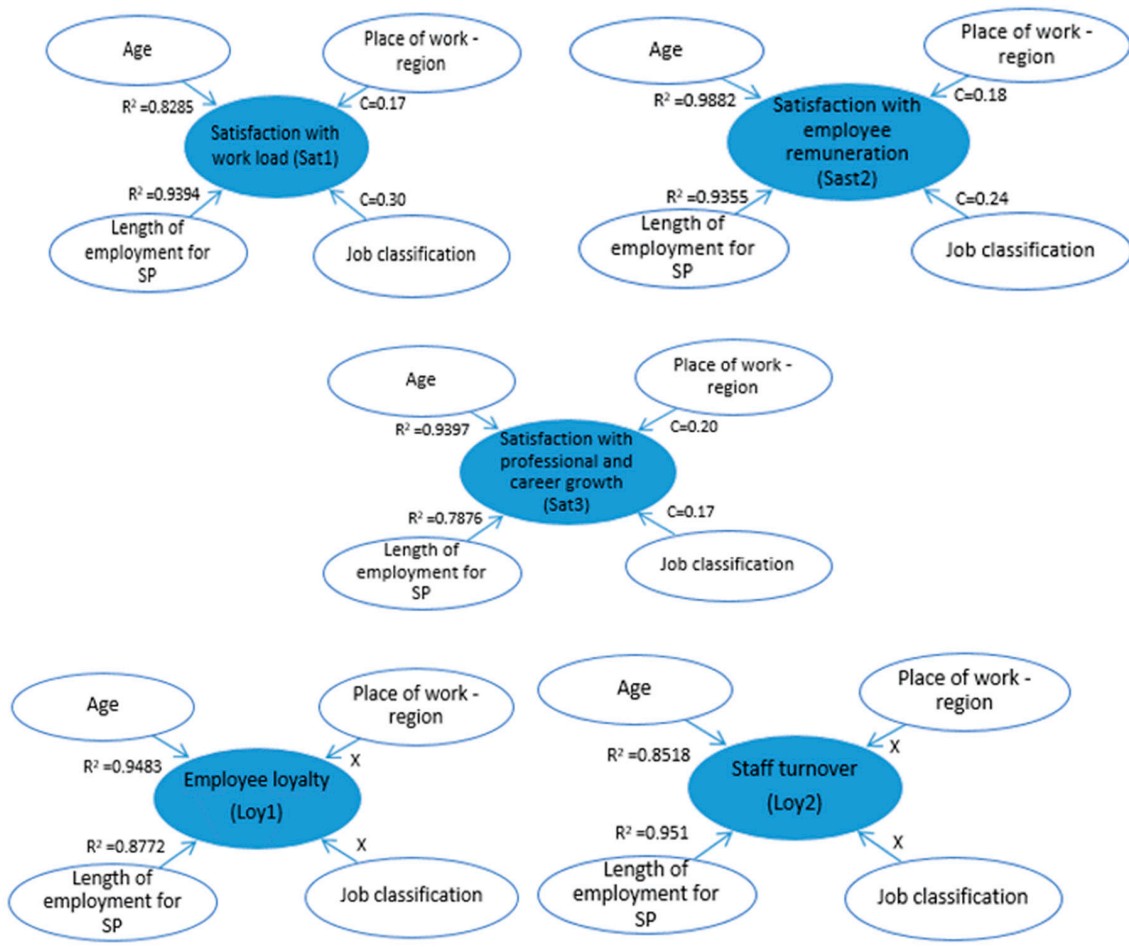

**Figure 9.** Impact of demographic characteristics to employee satisfaction and loyalty.

According to Lorincova et al. [17], Dirisu et al. [18], Razig and Maulabakhsh [19], Chinomona, and Dhurup [20], Bakotic and Babic [27], the quality of work life positively and significantly influences employee job satisfaction, and employee loyalty. The results of our research show that perceived quality (quality of workplace, quality of communications, quality of relationships, quality of teamwork, quality of education system) affect SP's employee satisfaction. Correlation coefficient 0.44 expresses the mean correlation. The value perceived by employees (assessing employee performance, evaluation of social services) and fulfilling of their expectations has a much greater impact on SP's employee satisfaction. Correlation coefficient 0.90 and 0.71 express very strong correlation. The fair evaluation of employee performance, social services, benefits and fulfilling their expectations in terms of rewarding, responsibility and workload, significantly affect the satisfaction of SP employees. The quality of working life is perceived by SP employees as a matter of course and they attach the highest priority to employee remuneration.

According to Chatzopoulou et al. [28], Kubala and Vetrakova [61], Sánchez-Sellero et al. [62], the employees satisfied with their reward and work environment do not have a need to leave the company; they are loyal. The results of our research show that the dependence between SP employee satisfaction and loyalty is strong (correlation coefficient 0.65). If SP wants to keep its employees, it is

necessary to increase their satisfaction. The focus should be on employee remuneration. Satisfaction rate with employee remuneration is very low (33.51%). The labor market analysis shows that the average nominal monthly wages of employees in the postal sector are long term below the average monthly wages of employees in the economy of the SR (Figure 1). Market opening and increased competition force Slovak Post to modernize their wage structure. Changes in minimum wages are an important driving force for salary development in the postal sector. If SP wants to increase loyalty and prevent employee fluctuation to competition, it needs to incorporate these facts into the creation of a new flexible employee remuneration system that takes into account these facts.

According to Giovanis [34], the positive causal effect from job classification on job satisfaction and employee loyalty is present. Our results partially confirmed this idea. The results of our examination of association confirmed the dependence between satisfaction of SP employees and job classification, however the level of relationship is low (Pearson coefficient 0.11–0.30). Based on the results of our research (Table 4) we can say that the most satisfied category of employees is technical and economics employees (satisfaction index 73.45%). This category is made up of the most loyal employees (79.16%). However, an examination of association confirmed the independence between loyalty and job classification.

Our results (Figure 8) are confirmed by the findings of survey by Kot-Radojewska and Timenko [35] who examined the relationship between employee loyalty to the employer and the form of employment. Their research results indicated that the employees who have an indefinite duration employment contract rated the degree of their own loyalty to the employer higher than people that have a fixed-term employment contract. We found out in our research, that the relationship between employment length and loyalty to SP has a slightly increasing linear progression; the determination coefficient value ($R^2 = 0.8772$) points to a strong relationship. This means that SP employees with longer experience (who have an indefinite employment contract) are more loyal than employees who work for a shorter period (predominantly have a fixed-term employment contract).

It is generally believed that job satisfaction increases linearly with age. The empirical analysis of the determinants of job satisfaction, provided by Clark et al. [38], confirmed that job satisfaction is U-shaped in relation to age. Our research has confirmed both claims. Examination of the dependences of satisfaction indicators on employee age (Figure 5) confirmed very strong relationships ($R^2 = 0.8285$–0.9882). The satisfaction with work load and the satisfaction with professional and career growth increase linearly with age. Satisfaction with employee remuneration is U-shaped in relation to age (polynomial course), declining from a moderate level in the younger employees and then increasing steadily up to retirement.

According to Gazioglu and Tansel [36], the employees in managerial, professional and clerical occupations are more satisfied than sales employees. Our research in conditions of a postal company (Slovak Post) confirmed this statement (Table 4). The employees with job classification "operation and sales", "workers" and "others" (subsidiary staff) are less satisfied than "technical and economics" employees (management and administration). Slovak Post has the biggest lack of workforce in operation and sales. This is due to the low satisfaction of employees in this category. Satisfaction with workload is 56.24% and satisfaction with remuneration is only 29.8%. Their work places high demands on professionalism, but wages are at the minimum wage level in Slovakia.

The topic of sustainable HRM has a broad scope. From creating appropriate working conditions, sustainable leadership, collaboration and teamwork, diversity and multiculturalism, ethics and governance, creating and inculcating values, health and safety, workforce engagement, employee training to environmental sustainability. Employee satisfaction and loyalty are key to making HRM sustainable. Slovak Post supports the idea for choosing sustainable HRM as a new approach for people management. The results of our research will used in the decision-making on future human resource development plans in Slovak Post.

## 6. Conclusions

The postal sector is a very labor intensive sector and national postal operators have traditionally been the largest domestic employers. The employment in the postal sector is influenced by the two main market developments: letter volumes decline and parcel volumes growth. Additionally, new technologies and growing competition, driven by the increase in parcel volumes and postal market liberalization, have also had an effect on postal employment volume and working conditions.

This fact has led to two key changes in the labor market.

1.  We observe an increase in new and more flexible employment models—on-call work, temporary agency work, performance-related pay contracts as well as outsourced models, such as sub-contracted workers and self-employment.

2.  We observe changes in employment conditions—increased competition has forced national postal operators (Slovak Post) to modernize their wage structure, e.g., introduce performance pay and other types of more flexible contracts [43].

The biggest threat to the postal services sector today is the lack of manpower, especially in the field of postal operation. This problem is caused by the low level of average monthly wage, which is not appealing for jobseekers, especially in developed regions. The top ten risk factors also includes the inability to attract, and also maintain, talent.

It is necessary that the focus of sustainable HRM of postal companies should be contributing not only to financial outcomes, but the importance of human, social and ecological outcomes in terms of their contribution to business outcomes. The postal operators implemented many activities related to sustainable human resource management, mainly concerning decent working conditions and employee training. Great emphasis is placed on eco-friendly solutions that facilitate, among other things, the work of employees. Quality working conditions, motivated and satisfied employees are the priority for postal operators. SP solves the human, social and individual questions SP in the social program. The main goal of SP's social program is to enhance the quality of life of its employees, work conditions and their overall personal development. SP adheres to an equal opportunities policy—the employer treats all employees equally. SP adopts measures against discrimination in all areas of the employment relationship and treats employees in accord with the principle of equal treatment. Employees are provided with professional education to enhance their qualifications in the area of innovation and unification of the SP product range and education in the area of information technology and foreign languages. SP has its environmental management system certified. SP exerts a maximum effort in the search for solutions that minimize the impact of its activities on the environment and successfully continues the process of the application of environmental management in the company. However, as our research has shown, it is also necessary to address employee remuneration issues. This will increase their satisfaction and loyalty, which will support the employer's attractiveness.

Sustainable HRM offers many opportunities for researchers from variety of disciplines and an opportunity to improve management practice. With our research, we have tried to zoom in on the issue of sustainable HRM in the postal sector. The added value of this investigation is the designed and applied HRSI model. The proposed HRSI model can be used as a basic diagnostic tool for monitoring employee satisfaction and loyalty. Its regular application enables postal companies to monitor the trend of development of individual latent variables and to identify weaknesses negatively affecting employee satisfaction and loyalty. The HRSI model has a general application in the conditions of any enterprise, but it is important to correctly select the measurable variables, the number of which depends on the depth of the investigation.

Our findings have practical implications for organizational leaders or HR professionals. Regular monitoring of employee satisfaction by applying the HRSI model will demonstrate the relevance of decisions taken on the basis of the results of previous measurements. Organizational leaders or HR professionals can follow the trend of satisfaction development and evaluate the effectiveness of decisions taken regarding work life of employees.

Employees' satisfaction enhancement is critical for the firm's sustainability and is the base of the company's sustained competitive advantage. The results of this study contribute to the sustainable HRM literature by application of a new diagnostic tool to increase the sustainable competitive advantage of postal company by making managerial decisions flexible in response to needs and expectations of employees and their satisfaction. Unlike the existing literature on sustainable HRM, we believe that our work can make a distinctive contribution to HRM research. The contribution for academics and professionals consists of mapping the development of employee satisfaction in the postal sector and comparing it with the results of other authors' research. The use of mathematical and statistical methods (regression and correlation analysis, association analysis) has contributed to a more accurate assessment of employee satisfaction based on socio-demographic characteristics.

It is necessary for postal operators to incorporate elements of sustainable HRM into their HR strategies, build upon the employer's brand, and become attractive employers who, in addition to financial remuneration, provide their employees with other benefits that outweigh the competitors. There is a need to strike a balance between free competition, consumer demands, the sustainability of the universal service and its funding, and the preservation of jobs. Employee satisfaction and employee loyalty form the basis of the sustainable HRM, and quality HRM has today had significant impacts on the future and sustainability of postal providers on the Slovak Postal market. Slovak Post, Human Resource Division, welcomed the cooperation and will use the results of our research in their decision-making on future human resource development plans.

**Author Contributions:** Conceptualization, M.S.; methodology, M.S.; validation, M.S. and K.A.; formal analysis, K.A.; investigation, M.S.; data curation, M.S. and K.A.; writing—original draft preparation, M.S.; writing—review and editing, M.S.; visualization, M.S.

**Funding:** This research was funded by projects VEGA 1/0518/19 Research of the digital economy development and its impact on the competitiveness of enterprises in the knowledge based society, and VEGA 1/0152/18 Business models and platforms in digital space.

**Acknowledgments:** This research was supported by projects VEGA 1/0152/18 Business models and platforms in digital space, and VEGA 1/0518/19 Research of the digital economy development and its impact on the competitiveness of enterprises in the knowledge based society.

**Conflicts of Interest:** The authors declare no conflict of interest.

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
