# Peer review of "Employee Satisfaction and Loyalty as a Part of Sustainable Human Resource Management in Postal Sector"

_sustainability, doi:10.3390/su11174591_

Round 1

Reviewer 1 Report

The text is based on an interesting idea and I really enjoyed reading it. The overall structure is good, but chapter 3 includes too many hypootheses and expectations - I feel that 4 - 5 would be more appropriate for this size of paper. I would also suggest transferring the hypotheses and expectations   from chapter 3 to chapter 4.

Author Response

Thank you for your opinion. I tried to complet all yours comments. Additionally, I modified the introduction, literature review.

The Discussion and the Conclusions have also been elaborated in more detail, including the evaluation of the measurement results, comparison with the literature and recommendations.

All changes are marked in red.

Thank you for cooperation.

Best regards

Mariana Strenitzerová

Detail corrections:

part 1. Introduction:

- modified

part 2. Sustainable human resource management - Literature review:

- modified,

- completed with a review about job satisfaction and loyalty

- added 2.1 Sustainable HRM in postal sector

part 3. Labor market situation in the EU and Slovak Republic postal sector:

- reduced

- hypotheses moved to part 4 (the number of hypotheses was not adjusted because of to evaluate them)

part 4. Materials and Methods:

- modified

- added hypotheses

part 5. Results:

+ results from part Discussion:

  5.2 Regression and correlation analysis of latent variables in HRSI model

5.3 Impact of demographic characteristics to employee satisfaction and loyalty

part 6. Discussion

- added

- evaluation of measurement results, comparison with literature and recommendations.

part 7. Conclusions

- modified, supplemented.

Reviewer 2 Report

The authors have chosen a topic regarding Sustainable HRM, employee satisfaction and loyalty. Specifically, they explore the impact of some factors on employee satisfaction and consequently on the loyalty of the employee.

Although, a priori, this idea is very interesting due to Sustainable HRM is underlined as an essential aspect in organizations (source of competitive advantage) and specially in postal sector; it should be emphasized that after its review there are some issues difficult of understanding.

The manuscript shows certain affirmations which, really, it would be very interesting to justify and reflectFor example (page 1, line 27) "The concept of sustainability is important for companies in the sector of postal servicesWhy? The explanation given below is not consistent with the justification of this sentence.

From the line 40, the link of the manuscript with the objective of the work is difficult to understand. The latent variables are not well justified in the theoretical part to build the model. It is very difficult to understand. Why are these factors (employer image, employee expectations, perceived HR service quality, value perceived by employees, etc.) that significantly affect the Sustainable HRM of postal companies?

The Sustainable HRM is a collection of references that are unrelated to each other and that do not lead to the objective of the manuscript. Finally, it ends up referring to the postal sector but it is disconnected and unrelated to the references previously shown.

It is recommended to read carefully and to reflect and justify the ideas exposed.

There is too much talk about the postal sector context and very little link to the topic of study. After the theoretical framework, the relationship between the study and the Sustainable HRM is no longer remembered.

There are hypothesis and expectations that are not located in the theoretical model. Why expectations? 

Methodology. The main issue of this article is the measurement of Sustainable HRM. How is measured this item? Are the factors or variables latent?

Minor issues: Review authors guidelines regarding references in the text.

It is recommended to increase the contribution in the conclusions.

It would be nice if authors could answer to:

What is the added value of this investigation?

What is the contribution for academics and professionals?

And, of course, what does it have to do with sustainability?

Author Response

Thank you for your opinion. I tried to complet all yours comments. Additionally, I modified the introduction, literature review.

The Discussion and the Conclusions have also been elaborated in more detail, including the evaluation of the measurement results, comparison with the literature and recommendations.

All changes are marked in red.

Thank you for cooperation.

Best regards

Mariana Strenitzerová

Detail corrections:

part 1. Introduction:

- modified

part 2. Sustainable human resource management - Literature review:

- modified,

- completed with a review about job satisfaction and loyalty

- added 2.1 Sustainable HRM in postal sector

part 3. Labor market situation in the EU and Slovak Republic postal sector:

- reduced

- hypotheses moved to part 4 (the number of hypotheses was not adjusted because of to evaluate them)

part 4. Materials and Methods:

- modified

- added hypotheses

part 5. Results:

+ results from part Discussion:

  5.2 Regression and correlation analysis of latent variables in HRSI model

5.3 Impact of demographic characteristics to employee satisfaction and loyalty

part 6. Discussion

- added

- evaluation of measurement results, comparison with literature and recommendations.

part 7. Conclusions

- modified, supplemented.

Your comments:

 Although, a priori, this idea is very interesting due to Sustainable HRM is underlined as an essential aspect in organizations (source of competitive advantage) and specially in postal sector; it should be emphasized that after its review there are some issues difficult of understanding.

The manuscript shows certain affirmations which, really, it would be very interesting to justify and reflectFor example (page 1, line 27) "The concept of sustainability is important for companies in the sector of postal servicesWhy? The explanation given below is not consistent with the justification of this sentence.

Added 2.1. Sustainable HRM in postal sector

“Postal companies become aware of their responsibilities towards employees and engage in sustainable human resource management. In particular, they consider the creation of the best working conditions for safety, health, social background and continuous training of employees as their role. Postal companies are making considerable efforts towards continuous improvement in the area of occupational safety and health protection. Equally in the implementation of education and in the consistent application of the principles of diversity and equal opportunities. An open communication culture, supporting employee engagement in these processes and activities, is a matter of course for postal companies. In this context, they realize that a motivating work environment is also important. All these activities aim at the main goal of postal companies in the area of human resources management - to increase the satisfaction and loyalty of their employees. The postal sector suffers from a lack of workforce and therefore postal companies pay great attention to the satisfaction and loyalty of employees as a part of sustainable human resource management. That is why Slovak post, as a universal postal service provider and the second largest employer in Slovakia, welcomed cooperation in solving our research.”

From the line 40, the link of the manuscript with the objective of the work is difficult to understand. The latent variables are not well justified in the theoretical part to build the model. It is very difficult to understand. Why are these factors (employer image, employee expectations, perceived HR service quality, value perceived by employees, etc.) that significantly affect the Sustainable HRM of postal companies?

Added in part 4.1 Employee job satisfaction – methodology

“When creating the HRSI model, we were based on the general ECSI (European Customer Satisfaction Index) model, in which we considered Slovak post employees as an internal customer of the Human Resources Division of the Slovak post. The ECSI model monitors seven areas (latent variables) that have a determining impact on customer satisfaction, namely product/service/company image, customer expectations, customer perception, customer perceived value, customer satisfaction, customer complaints and customer loyalty. Measurable variables (indicators) within the monitored areas were determined after consultation with the management of Human Resources Division.”

 The Sustainable HRM is a collection of references that are unrelated to each other and that do not lead to the objective of the manuscript. Finally, it ends up referring to the postal sector but it is disconnected and unrelated to the references previously shown. It is recommended to read carefully and to reflect and justify the ideas exposed.

Modified, completed with a review about job satisfaction and loyalty

 There is too much talk about the postal sector context and very little link to the topic of study. After the theoretical framework, the relationship between the study and the Sustainable HRM is no longer remembered.

-          reduced

  “The analysis of the labor market situation in the postal sector, the development of the number of employees and the development of the average wage in the postal sector have served as a basis for the assessment of employee satisfaction with the remuneration system and for the survey of employee loyalty.”

There are hypothesis and expectations that are not located in the theoretical model. Why expectations? 

The validity of the hypotheses has been tested and the validity of the expectations has not been tested, that result is valid only for sample.

Minor issues: Review authors guidelines regarding references in the text.

corrected

It is recommended to increase the contribution in the conclusions.

It would be nice if authors could answer to:

What is the added value of this investigation?

What is the contribution for academics and professionals?

And, of course, what does it have to do with sustainability?

Added in part Discussion and Conclusion

“Sustainable Human Resource Management offers many opportunities for researchers from variety of disciplines and an opportunity to improve management practice. With our research, we have tried to zoom in the issue of sustainable human resources management in the postal sector. The added value of this investigation is the designed and applied HRSI model, which can be of general use. Contribution for academics consists in mapping the development of employee satisfaction in the postal sector and comparing it with the results of other authors' research. The use of mathematical and statistical methods has contributed to a more accurate assessment of employee satisfaction based on socio-demographic characteristics. Slovak Post, Human Resources Division welcomed the cooperation and will use the results of our research in their decision-making on future human resource development plans.”

Reviewer 3 Report

Review Report

According to the authors, the study “proposes a new perspective of employee satisfaction assessment that not only quantifies total satisfaction but identifies job attributes and socio-demographic characteristics affecting employee satisfaction and loyalty as a key concern for sustainable human resource management

The influence of human resources on organizational sustainability has been largely analyzed in literature, approaching several different issues, such as innovation. In this context, as reported by Authors, considered as an extension of strategic HRM, sustainable HRM proposes a new approach to people management with the focus on long-term human resource development, regeneration, and renewal. Overall, I think that the topic approached in this paper fits well in the theme of this journal.

Nevertheless, although I think that the paper may rise to the standards expected by the Sustainability journal, I have several concerns and questions that, as a reviewer I would like to bring up:

·       First of all, in the Introduction and the Conclusions sections, contributions to theory and practice are not clear. Authors need to enlighten readers about these contributions.

·       The literature review and hypotheses development section needs to be extensively revised. Several papers on sustainable HRM have been published during last years. Authors need to present a proper review concerning this core theme. Moreover, Authors present a total of 10 hypotheses, but the paper lacks a proper hypotheses development section, through proposing relevant arguments which could sustain each of the proposed hypotheses. For such a purpose, Authors may reduce their section 3 “Labor market situation in the EU and Slovak Republic postal sector” (which seems too large comparing to other more important section), and thus avoiding extending too much the global size of the paper.

·      Concerning the methodological section, Authors need to provide additional insights concerning the measurement of each latent variables. Why did Authors used these measurable variables summarized in table 5, and not others? What literature sustains these options? Authors refer that “Factor analysis was used to summarize indicators (measurable variables) which identify the crucial factors”. This is not sufficient for readers.

·       Moreover, when self-report questionnaires are used to collect data, common method bias (CMB) may be a concern. I would like to see in this paper how authors addressed this issue. The work of Chang et al. (2010), Podsakoff (2003), Dillman's (2000) and the NRC (2013) might help. Authors should also consider the use of the Harman's (1967) single factor test to examine the likelihood of common method bias threat.

·       Concerning data analysis, and considering Authors’ objectives and model, why didn’t they apply structural equation modelling procedures. Results would have been stronger.

·       Authors propose a section 6, called “Discussion”, but the truth is that this section just brings more results. Thus, Authors should move the content of this section to the Results section (re-organizing it) and should provide a proper discussion section, where results are analysed in the light of theory.

Good work!

Author Response

Thank you for your opinion. I tried to complet all yours comments. Additionally, I modified the introduction, literature review.

The Discussion and the Conclusions have also been elaborated in more detail, including the evaluation of the measurement results, comparison with the literature and recommendations.

All changes are marked in red.

Thank you for cooperation.

Best regards

Mariana Strenitzerová

Detail corrections:

part 1. Introduction:

- modified

part 2. Sustainable human resource management - Literature review:

- modified,

- completed with a review about job satisfaction and loyalty

- added 2.1 Sustainable HRM in postal sector

part 3. Labor market situation in the EU and Slovak Republic postal sector:

- reduced

- hypotheses moved to part 4 (the number of hypotheses was not adjusted because of to evaluate them)

part 4. Materials and Methods:

- modified

- added hypotheses

part 5. Results:

+ results from part Discussion:

  5.2 Regression and correlation analysis of latent variables in HRSI model

5.3 Impact of demographic characteristics to employee satisfaction and loyalty

part 6. Discussion

- added

- evaluation of measurement results, comparison with literature and recommendations.

part 7. Conclusions

- modified, supplemented.

Your comments:

First of all, in the Introduction and the Conclusions sections, contributions to theory and practice are not clear. Authors need to enlighten readers about these contributions.

part 1. Introduction:

- modified

part 6. Discussion

- added

- evaluation of measurement results, comparison with literature and recommendations.

part 7. Conclusions

- modified, supplemented.

·       The literature review and hypotheses development section needs to be extensively revised. Several papers on sustainable HRM have been published during last years. Authors need to present a proper review concerning this core theme. Moreover, Authors present a total of 10 hypotheses, but the paper lacks a proper hypotheses development section, through proposing relevant arguments which could sustain each of the proposed hypotheses.

part 2. Sustainable human resource management - Literature review:

- modified,

- completed with a review about job satisfaction and loyalty

- was added the relevant arguments which could sustain each of the proposed hypotheses.

- added 2.1 Sustainable HRM in postal sector

For such a purpose, Authors may reduce their section 3 “Labor market situation in the EU and Slovak Republic postal sector” (which seems too large comparing to other more important section), and thus avoiding extending too much the global size of the paper.

reduced

“The analysis of the labor market situation in the postal sector, the development of the number of employees and the development of the average wage in the postal sector have served as a basis for the assessment of employee satisfaction with the remuneration system and for the survey of employee loyalty.”

·      Concerning the methodological section, Authors need to provide additional insights concerning the measurement of each latent variables. Why did Authors used these measurable variables summarized in table 5, and not others? What literature sustains these options? Authors refer that “Factor analysis was used to summarize indicators (measurable variables) which identify the crucial factors”. This is not sufficient for readers.

Added in part 4.1 Employee job satisfaction – methodology

“When creating the HRSI model, we were based on the general ECSI (European Customer Satisfaction Index) model, in which we considered Slovak post employees as an internal customer of the Human Resources Division of the Slovak post. The ECSI model monitors seven areas (latent variables) that have a determining impact on customer satisfaction, namely product/service/company image, customer expectations, customer perception, customer perceived value, customer satisfaction, customer complaints and customer loyalty. Measurable variables (indicators) within the monitored areas were determined after consultation with the management of Human Resources Division.”

·       Moreover, when self-report questionnaires are used to collect data, common method bias (CMB) may be a concern. I would like to see in this paper how authors addressed this issue. The work of Chang et al. (2010), Podsakoff (2003), Dillman's (2000) and the NRC (2013) might help. Authors should also consider the use of the Harman's (1967) single factor test to examine the likelihood of common method bias threat.

Procedural remedy:

Data collection: separate independent and dependant variable data collection in different timing (physical separation);

Psychological separation: ....

Scale: ...

Grouping items in questionnaire:

We did not use Harman's one-factor test to examine the amount of biasness. It was not our aim.

·       Concerning data analysis, and considering Authors’ objectives and model, why didn’t they apply structural equation modelling procedures. Results would have been stronger.

Structural equation modelling is a much more complicated system for analysis a multiple relationships. We used the simple linear regression analysis mainly for its simplicity and moreover requirements of submitter did not require deeper analysis. Human Resources Division demanded results in a simple, readable form, understandable to both management and staff.

·       Authors propose a section 6, called “Discussion”, but the truth is that this section just brings more results. Thus, Authors should move the content of this section to the Results section (re-organizing it) and should provide a proper discussion section, where results are analysed in the light of theory.

Move to the Results: Regression and correlation analysis of latent variables in HRSI model, Impact of demographic characteristics to employee satisfaction and loyalty

Discussion: added

- evaluation of measurement results, comparison with literature and recommendations.

Round 2

Reviewer 2 Report

With the changes made, I identify that the paper has been improved significantly considering all the previous comments.

The writing of the theoretical framework makes it more understandable to the reader. The variable description and the contribution are much clearer than the last manuscript.

Minor issues: Change the hypotheses to the theoretical part.

Congratulations for the improvements made.

Author Response

Thank you for your opinion. I tried to complet all yours comments.

All changes are marked in blue.

Thank you for cooperation.

Best regards

Mariana Strenitzerová

Your comments:

Change the hypotheses to the theoretical part:

I changed the chapter 2 and added Section 2.4. Hypotheses development:

Theoretical framework and hypotheses development

2.1. Sustainable human resource management literature

2.2. Sustainable HRM in postal sector

2.3. Labor market situation in the EU and Slovak Republic postal sector

2.4. Hypotheses development

Reviewer 3 Report

Review Report

The paper has improved significantly. Authors have addressed several of my previous comments and suggestions. Nevertheless, although I think that the paper may rise to the standards expected by the Sustainability journal, I still have few concerns and questions that, as a reviewer I would like to bring up:

·       Authors have added some considerations about workforce in the Postal sector in the introduction section and reformulated their conclusions section, but in the Introduction and the Conclusions sections, contributions to theory and practice are still not clear. Authors need to enlighten readers about these contributions. How does this paper can extend the literature body on SHRM? How can professionals may benefit from these results?

·       The literature review and hypotheses development section was properly extended. However, the paper still lacks a proper hypotheses development section, through proposing relevant arguments which could sustain each of the proposed hypotheses.  In accordance, my suggestion is changing the title “2. Sustainable human resource management – Literature review” to Theoretical framework and hypotheses development”. Then, the section would begin with a section called “2.1. Sustainable human resource management literature”. Then, “2.2. Sustainable HRM in postal sector”, “2.3. Labor market situation in the EU and Slovak Republic postal sector”, and finally “2.4. Hypotheses development”. In the “2.4. Hypotheses development” section, author needs to provide valid arguments concerning EACH hypothesis in a sequential approach. Moreover, expectation 1 is not clear: E1: The greatest impact on employee satisfaction has perceived quality of HR services – what does this means?

·       Authors referred that they did not use Harman's one-factor test to examine the amount of biasness, because It was not their aim, but this is not a question of aim, but a way to provide sufficient evidences concerning the presence or not of common method bias, which is important to trust in results achieved.

Good work!

Author Response

Thank you for your opinion. I tried to complet all yours comments.

All changes are marked in blue.

Thank you for cooperation.

Best regards

Mariana Strenitzerová

Your comments:

Authors have added some considerations about workforce in the Postal sector in the introduction section and reformulated their conclusions section, but in the Introduction and the Conclusions sections, contributions to theory and practice are still not clear. Authors need to enlighten readers about these contributions. How does this paper can extend the literature body on SHRM? How can professionals may benefit from these results?

incorporated in the introduction and conclusion

The literature review and hypotheses development section was properly extended. However, the paper still lacks a proper hypotheses development section, through proposing relevant arguments which could sustain each of the proposed hypotheses.  In accordance, my suggestion is changing the title “2. Sustainable human resource management – Literature review” to Theoretical framework and hypotheses development”. Then, the section would begin with a section called “2.1. Sustainable human resource management literature”. Then, “2.2. Sustainable HRM in postal sector”, “2.3. Labor market situation in the EU and Slovak Republic postal sector”, and finally “2.4. Hypotheses development”. In the “2.4. Hypotheses development” section, author needs to provide valid arguments concerning EACH hypothesis in a sequential approach. Moreover, expectation 1 is not clear: E1: The greatest impact on employee satisfaction has perceived quality of HR services – what does this means?

I changed the chapter 2 according to your suggestion and added Section 2.4. Hypotheses development.

The expectation E1: The greatest impact on employee satisfaction has the perceived quality of HR services: We suppose, that the quality of working life, the creation of a quality working environment, the quality of personnel activities, the personnel policy of SP, the quality of services of Human Resources Division provided to employees will positively influence the satisfaction of SP employees.

Authors referred that they did not use Harman's one-factor test to examine the amount of biasness, because It was not their aim, but this is not a question of aim, but a way to provide sufficient evidences concerning the presence or not of common method bias, which is important to trust in results achieved.

Thank you very much for the idea of applying Harman's one-factor test to examine the amount of biasness. In the future, we plan to repeat the research to monitor the development of the measurable variables of the HRSI model. Slovak Post is interested in cooperation in the future. When evaluating the results, we will apply the Harman's one-factor test and publish its results.

Round 3

Reviewer 3 Report

Although some of the recommendations were not followed (or not enough), the manuscript has improved a lot, and I think that the paper may be accepted for publication.

Good work.